# CasTuner is a degron and CRISPR/Cas-based toolkit for analog tuning of endogenous gene expression

Gemma Noviello ®[1], Rutger A. F. Gjaltema[1,2] & Edda G. Schulz ®[1] ✉

Certain cellular processes are dose-dependent, requiring specific quantities or stoichiometries of gene products, as exemplified by haploinsufficiency and sex-chromosome dosage compensation. Understanding dosage-sensitive processes requires tools to quantitatively modulate protein abundance. Here we present CasTuner, a CRISPR-based toolkit for analog tuning of endogenous gene expression. The system exploits Cas-derived repressors that are quantitatively tuned by ligand titration through a FKBP12$^{F36V}$ degron domain. CasTuner can be applied at the transcriptional or post-transcriptional level using a histone deacetylase (hHDAC4) fused to dCas9, or the RNA-targeting CasRx, respectively. We demonstrate analog tuning of gene expression homogeneously across cells in mouse and human cells, as opposed to KRAB-dependent CRISPR-interference systems, which exhibit digital repression. Finally, we quantify the system's dynamics and use it to measure dose-response relationships of NANOG and OCT4 with their target genes and with the cellular phenotype. CasTuner thus provides an easy-to-implement tool to study dose-responsive processes in their physiological context.

Biological processes are often dose-dependent, meaning that they rely not only on the presence or absence, but on defined quantities of specific RNAs or proteins. Such dose sensitivity can arise from the need to maintain the right stoichiometry within a protein complex[1]. It might also evolve to restrict a process to a certain cell type or spatial position within an embryo (e.g. by sensing a morphogen gradient)[2]. As a consequence, a subset of genes exhibit haploinsufficiency[1,3] and a dedicated process has evolved in many species to ensure dosage compensation for X-linked genes between the sexes[4].

The process responsible for X-dosage compensation in mammals, X-chromosome inactivation, is itself controlled in a dosage-sensitive manner. It is restricted to female cells by sensing the two-fold higher dose for X-linked genes in females compared to males[5,6]. Another example for a gene-dosage sensitive process is the differentiation of pluripotent stem cells into different lineages. Here, relatively small variations in the amount of the pluripotency factor OCT4 (POU5F1) can determine whether mouse embryonic stem cells (mESCs) remain in the pluripotent state or differentiate into trophectoderm or meso-endoderm lineages[7,8]. Similarly, the precise quantity of the pluripotency factor NANOG is critical for the control of naive and primed pluripotent states both in vitro and in vivo[9,10]. Understanding the principles underlying dose-dependent regulation of biological processes is, thus, of critical importance. It is however technically challenging, since it requires the ability to quantitatively modulate protein abundance.

The first systems developed that could potentially allow quantitative control of gene expression were inducible promoters such as the TetON/OFF system controlling overexpression of a gene from cDNA[11]. However, these systems typically overexpress genes beyond their physiological levels and often show uninduced or leaky expression[12,13]. Moreover, intermediate expression can be difficult to achieve at the single-cell level[14]. Although more complex circuits have been designed to improve quantitative control of gene expression (tunability)[15], recent technological developments, such as conditional destabilising

---

[1]Systems Epigenetics, Otto Warburg Laboratories, Max Planck Institute for Molecular Genetics, 14195 Berlin, Germany. [2]Present address: Swammerdam Institute for Life Sciences, University of Amsterdam, Science Park 904, 1098 XH Amsterdam, The Netherlands. ✉e-mail: edda.schulz@molgen.mpg.de

domains (degrons) and Cas9-based approaches seem to be more promising for tuning protein abundance[16–18]. Since they allow control of endogenous genes, they can operate at physiological expression levels.

CRISPR/Cas9-based epigenome editing relies on a catalytically dead Cas9 (dCas9) fused to an effector domain, which is then targeted to a gene promoter through a single guide RNA (sgRNA) to either repress (CRISPRi)[19] or activate (CRISPRa) a target gene[20]. A more recent alternative are RNA-targeting CRISPR systems, such as CasRx (Cas13d)[21]. Different approaches have been applied to make CRISPRa/i inducible, typically relying on conditional stabilisation or on inducible dimerisation between dCas9 and the effector domain[22]. A disadvantage of the latter approach is a partial gene repression in the absence of effector domain recruitment, because dCas9 itself still occupies the target gene promoter[19]. Therefore, conditional stabilisation is the preferred option for CRISPRi, where a conditional degron domain is fused to a protein of interest to alter its stability in response to stimuli such as binding of ligands[18]. Although a series of studies have tested different designs of degron-controlled dCas9 systems, they have mostly been applied to CRISPRa and have not investigated tunability at the single-cell level[17,23–25]. Moreover, the most widely used repressor domain, KRAB, has been shown to operate in a switch-like manner[26], which would make this classical CRISPRi system unsuitable to generate homogeneous intermediate expression levels. Since in particular CRISPRi would allow modulating expression levels in a physiological meaningful range, we sought to develop a CRISPR/Cas-based system that is tunable at the single-cell level by varying the concentration of a ligand.

We tested a panel of degron and repressor domains and identified two designs that supported potent and tunable repression of a fluorescently-tagged endogenous gene in mESCs. In this system, which we named CasTuner, a FKBP12$^{F36V}$ degron domain[27] controls dCas9 fused to a human histone deacetylase 4 (hHDAC4) repressor domain[28] or the RNA-targeting CasRx protein[21]. For both systems we quantify the dynamic properties of gene repression and derepression. Our data show that, by titrating repressor abundance using different concentrations of ligand, CasTuner can quantitatively perturb a gene of interest within physiologically relevant ranges homogeneously across cells. We have thus designed a toolkit that allows rapid, inducible, tunable and reversible gene repression at the transcriptional or posttranscriptional level. We show applications of CasTuner in mouse embryonic stem cells and in human HeLa cells. Moreover, we demonstrate the applicability of the system in studying the dose-dependent action of NANOG and OCT4 in controlling target gene expression and cellular phenotype.

## Results

### The AID and FKBP12$^{F36V}$ degron domains allow potent control of dCas9 abundance

We sought to create a tool for tuning endogenous gene expression that can be easily applied to any target gene. We reasoned that, by fusing a CRISPR-based artificial repressor with a conditional degron domain, we can titrate its quantity and thereby titrate the expression of a target gene. In the first steps, we aimed at identifying suitable degron and repressor domains.

An ideal degron domain would support a wide dynamic range of repressor levels and its complete removal, when not needed. To compare different degrons, we expressed dCas9 fused to a red fluorescent protein (tRFP) tagged with different degron domains, followed by a P2A site and a blue fluorescent protein (tBFP). In these constructs, dCas9-tRFP and tBFP are transcribed together as a bicistronic unit, but are then translated into two separate proteins (Fig. 1a, top). The effect of the degron domain can be quantified as the ratio between tRFP, as a proxy for dCas9 levels, and tBFP, which should remain constant. Using this platform, we evaluated the performance of 6 different degron

domains, each fused N-terminally to dCas9: AID[29], mAID[30], SMASh[31], FKBP12$^{F36V}$[27], ecDHFR[32] and ER50[33]. As a control we used the same dCas9 construct without any degron (no-degron control) (Fig. 1a). All constructs were stably integrated in mESCs through PiggyBac (PB) transposition and a cell population with homogeneous tBFP expression was generated by fluorescence-activated cell sorting (FACS) (Fig. 1a, bottom). For the AID-dCas9 and mAID-dCas9 constructs, OsTIR1, an accessory protein required for Auxin-induced proteasomal degradation is expressed from the same construct within the resistance cassette (Supplementary Fig. 1a).

We divided the degrons into two groups based on the change in stability of the fusion protein upon ligand addition (Fig. 1b, top): degrons that induce degradation (default stable) and degrons that induce stabilisation (default unstable). We treated each degron-dCas9 cell line with 4 concentrations of degron-specific ligand for 24 h and measured tRFP and tBFP levels by flow cytometry (Fig. 1b–d and Supplementary Fig. 1b–g). To account for non-specific ligand effects, we also tested the highest concentration of each ligand and the mock treatment on the no-degron control line. For each degron we analysed three different properties (Fig. 1b, bottom): the degradation leakiness, as a measure of destabilisation in the absence of induced degradation (Fig. 1e), the degradation efficiency, which quantifies the destabilisation upon maximal degradation (Fig. 1f), and the dynamic range of the system, describing the fold change between the stabilised and destabilised conditions (Fig. 1g).

Only two out of the 6 tested degron domains were able to efficiently control dCas9 levels, namely the Auxin-controlled AID system and the dTAG-controlled FKBP12$^{F36V}$ degron. Both exhibited a wide dynamic range (43–45 fold change, Fig. 1g), very high degradation efficiency (97–98%, Fig. 1f) and intermediate degradation leakiness (31% and 20%, Fig. 1e). Since FKBP12$^{F36V}$ showed a slightly higher fold change and a lower degradation leakiness than AID, it was employed in all subsequent experiments for post-translational control of Cas-repressor systems.

### The FKBP12$^{F36V}$ degron domain can reliably control Cas-mediated repression of an endogenous gene

Previous work had suggested that the widely used KRAB repressor domain acts in a switch-like manner[26] and might therefore not be suitable for quantitative control of gene expression. Thus, we included two additional repression systems in our analyses and compared efficiency, dynamics and homogeneity of repression. We chose dCas9 fused to a histone deacetylase (hHDAC4), which has been shown previously to enable potent repression[26], and the RNA-targeting CasRx[21]. While the KRAB domain induces histone H3 lysine 9 trimethylation (H3K9me3), hHDAC4 catalyses histone deacetylation[28,34] and CasRx leads to RNA degradation[21] (Fig. 2a). For the dCas9-repressor systems we tested both N- and C-terminal fusion of the repressor domain. All constructs were tagged with the FKBP12$^{F36V}$ degron at their N-terminus and with tBFP at their C-terminus, which allowed monitoring of repressor levels by flow cytometry. For the KRAB-repressor, we created an additional construct, where dCas9 and KRAB are not directly fused, but tethered by the ABA-inducible PYL1/ABI dimerisation system (here referred to as KRAB-Split-dCas9)[35], with which we have achieved potent repression in the past[36].

To compare their ability to tune endogenous gene expression, we targeted the repressor systems to the *Esrrb* gene in a mESC line, where the gene is homozygously tagged with P2A-mCherry (Fig. 2a)[36]. We then used flow cytometry to quantify repression at the single-cell level. Previous work had shown that mESCs lacking *Esrrb* retain self-renewal and appear morphologically normal[37], making it a suitable reporter system to test different repression mechanisms on endogenous gene expression.

For each repressor construct, we created stable cell lines through PB transposition followed by two consecutive rounds of cell sorting

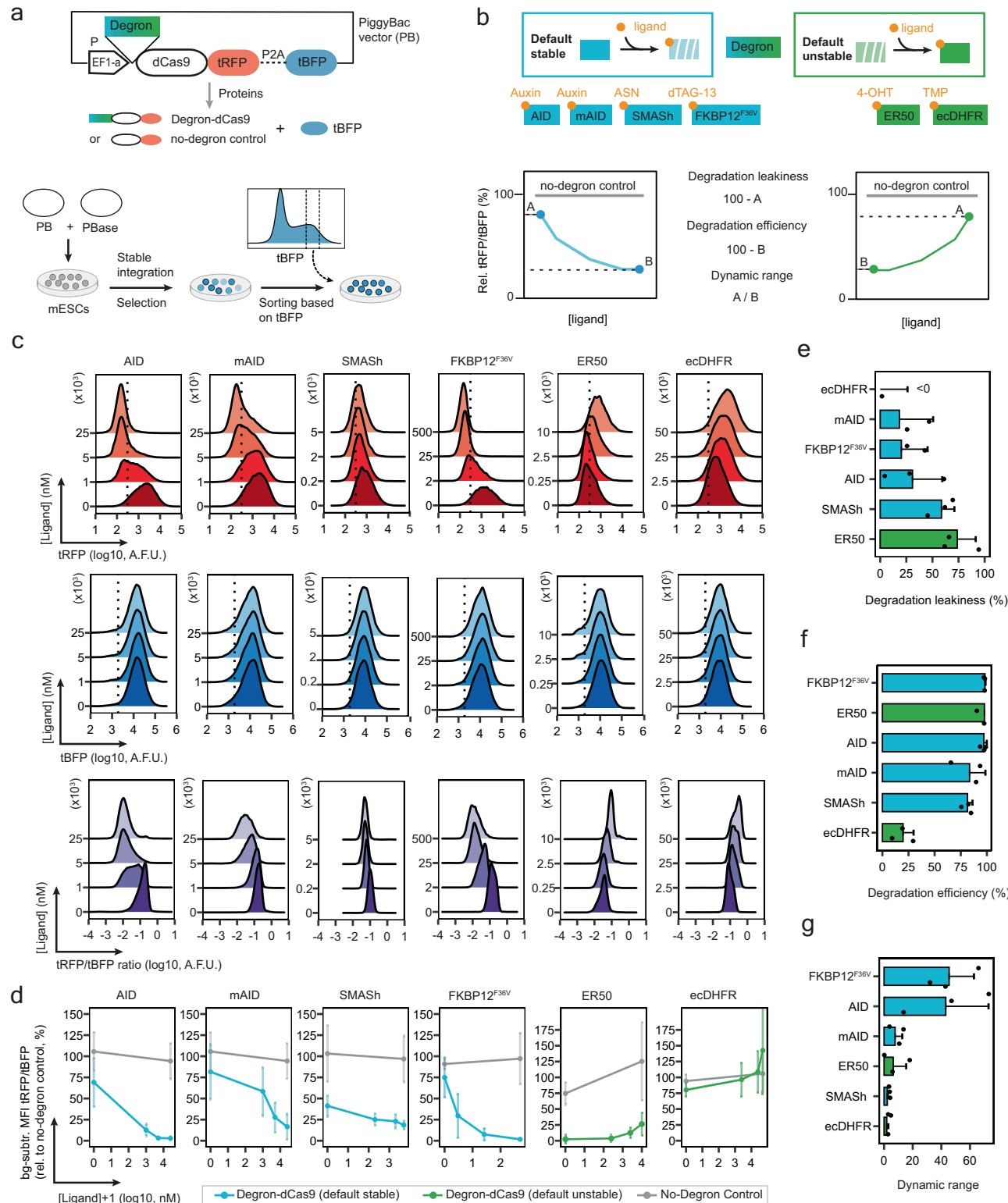

based on tBFP levels, in order to obtain cells with homogeneous repressor abundance (Fig. 2b). We then transduced cells with a sgRNA vector, co-expressing three different guide RNAs targeting *Esrrb* or expressing non-targeting control (NTC) guides. Transduction was performed in the absence of the targetable repressor protein (high dTAG-13). To induce repression, the dTAG-13 degrader was withdrawn for 4 days (Fig. 2b). We then measured ESRRB expression (mCherry) and the quantity of repressor (tBFP) by flow cytometry (Fig. 2c–g and Supplementary Fig. 2).

All degron-Cas-repressors were efficiently degraded with the exception of the C-terminal hHDAC4 construct (dCas9-hHDAC4), where tBFP levels were only reduced by ~56% (Fig. 2c–d). When comparing mCherry levels in the presence and absence of dTAG-13, the N-terminal fusion of the repressor domain resulted in strong depletion for both KRAB and hHDAC4 fusions (-83–89%), which was somewhat weaker for the C-terminal fusion constructs (-80%) (Fig. 2e,f). CasRx also induced a clear knock-down with 79% mCherry reduction. The strength of repression did not strictly depend on the repressor level,

**Fig. 1 | Comparison of degron domains to control dCas9 stability. a** Schematic of PiggyBac (PB) plasmids used to assess degron functionality. dCas9 is fused to tagRFP-T (tRFP), followed by a P2A peptide and tagBFP (tBFP) and expressed under the control of the EF1-alpha promoter. A degron domain is fused N-terminally to dCas9, while in the "no-degron control" dCas9 is not degron-tagged. The P2A peptide allows cleavage of (degron-)dCas9-tRFP and tBFP into two separate proteins. The PB plasmid is cotransfected with a hyperactive PB transposase (PBase) into mESCs (Tx1072 line). Cells that have genomically integrated the PB plasmid in (possibly multiple) genomic locations are selected with blasticidin and then sorted by FACS based on their tBFP level with identical gates for all constructs, yielding a homogenous population of cells. **b** Schematic overview of degron-dCas9 comparison as shown in (**c–g**). Top: overview over the tested degrons, their ligands (orange) and whether they are destabilised (default stable, blue) or stabilised (default unstable, green) by ligand addition. Bottom: based on the tRFP/tBFP ratios measured without ligand, at maximal ligand concentration and for the no-degron control, three parameters (see Methods for details) are estimated to characterise the ability to control dCas9 for each degron: degradation leakiness (as measure of the minimal destabilisation conferred by the degron), the degradation efficiency (maximal destabilisation) and the dynamic range (maximal fold change). **c** Density plots for tRFP (top), tBFP (middle) and the tRFP/tBFP ratio (bottom) for one biological replicate after 24 h of treatment. Dotted lines show the 99th percentile of non-fluorescent control cells. A.F.U. = Arbitrary Fluorescence Units. **d** tRFP-to-tBFP ratio, calculated as Median Fluorescence Intensity (MFI) after subtraction of the background fluorescence. Dots represent the mean of three biological replicates and are connected by lines; vertical lines indicate the s.d. **e–g** Bar plots of mean degradation leakiness (**e**), degradation efficiency (**f**) and dynamic range (**g**) calculated for the different degrons as depicted in (**b**). Degron domains from top to bottom are ranked from best to worst for each property. Black dots are single replicates. Error bars indicate the s.d. of three biological replicates. bg-subtr. = background-subtracted.

since FKBP12$^{F36V}$-dCas9-hHDAC4 was expressed at higher levels compared to FKBP12$^{F36V}$-hHDAC4-dCas9, but induced less repression (Supplementary Fig. 2f). To assess whether residual degron-Cas-repressor protein in the presence of dTAG-13 would result in unwanted repression (background silencing), we compared mCherry levels between cells transduced with a targeting and a non-targeting sgRNA construct. We could not detect any significant difference between the two lines, which showed that repression was effectively prevented by repressor degradation and no repression leakiness was observed (Fig. 2g).

In summary, we have generated a set of degron-controlled Cas-repressor constructs that can efficiently repress a target in a strictly inducible manner. We found an optimised design, where the repressor domain is fused N-terminally to dCas9, which confers increased repressor activity and will be characterised further in the next sections.

## HDAC4-dCas9 and CasRx-mediated repression allows analog tuning of gene activity

Having optimised several degron-Cas-repressor systems, we next aimed at comparing their ability to tune endogenous gene expression at the single-cell level. We refer to repression "tunability" as the ability to partially repress a target gene, resulting in stable intermediate expression levels. Our goal is to homogeneously titrate protein abundance by controlling the quantity of repressor. Depending on the repression mechanism, intermediate repressor levels could in principle lead to two alternative outcomes: a gradual and homogenous change in target gene expression or a bimodal pattern with a positive and a negative population of cells. Hence, the modality of repression can be defined as "analog" in the first scenario and "digital" in the second (Fig. 3a).

We used our endogenous Esrrb-mCherry reporter system and titrated each degron-Cas-repressor through treatment with varying dTAG-13 concentrations. We characterised CasRx and the dCas9 constructs with N-terminal repressor domains, but also the effect of dCas9 alone (KRAB-Split-dCas9 without ABA treatment), which likely acts through steric hindrance of transcription.

By using a range of 12 dTAG-13 concentrations, we were able to homogeneously titrate protein abundance (tBFP) of all 5 repressors (Fig. 3b, Supplementary Fig. 3). When analysing target gene expression (Fig. 3c,d), we observed that KRAB-mediated repression was effective at lower repressor levels than hHDAC4 and CasRx (Fig. 3e). The KRAB domain thus appears to be the most potent repressor of the three. When analysing tunability at intermediate repressor levels we observed two different repression patterns: CasRx, dCas9 alone and dCas9-hHDAC4 exhibited gradual tuning of mCherry levels, while KRAB-mediated repression gave rise to a bimodal distribution (Fig. 3c, Supplementary Fig. 3f, Hall-York test for unimodality, $p < 0.01$, see Methods for details). Repression tunability can be verified only by single-cell measurements, since at the cell population level, bimodal or homogeneous intermediate distributions are indistinguishable (Fig. 3c vs 3d). Comparison of mCherry-high and -low cells at intermediate KRAB repression strength revealed that both populations expressed similar amounts of tBFP (Fig. 3f). The observed bimodal repression pattern is thus unlikely to arise from heterogeneity in repressor levels, but appears to be an inherent property of the repression mechanism. For CasRx and to a lesser extent, for dCas9-hHDAC4, by contrast, mCherry and tBFP levels were negatively correlated, suggesting that cell-to-cell variability in repressor levels results in variable repression strength (Fig. 3f). This effect could potentially be reduced by generating clonal cell lines with more uniform repressor levels. Among the tested systems, only hHDAC4-dCas9 and CasRx are able to tune gene expression at the single-cell level. Therefore, we named these two systems collectively the CasTuner toolkit.

## Speed and reversibility of repression in degron-Cas-repressor systems

Having identified several degron-Cas-repressor designs that can quantitatively tune gene expression, we set out to further characterise their dynamics and the reversibility of repression, using our endogenous Esrrb-mCherry mESC lines. In addition to the tunable CasRx and hHDAC4-dCas9 systems of the CasTuner toolkit we again included the widely used KRAB systems and dCas9 alone for comparison. The dynamics of repression upon ligand withdrawal and derepression upon ligand addition depend on the repressor dynamics, but also on the speed with which the repressed state is established and erased, respectively. For CasRx we expect establishment and erasure of repression to be immediate, since here the repressor itself will cleave the mRNA. For hHDAC4 and KRAB-dependent systems these processes might be slower, since they affect gene expression in a less direct manner through modifying the chromatin state at the gene promoter.

To assess the repression and derepression dynamics, we induced repressor upregulation in Esrrb-mCherry mESCs by dTAG-13 withdrawal and repressor degradation by dTAG-13 addition, respectively. In each case we then monitored repressor (tBFP) and target gene (mCherry) levels at different time points over 6 days (Fig. 4, Supplementary Fig. 4a–d). To disentangle the different steps that control the system's dynamics, we parameterised ordinary differential equation (ODE) models of the system based on the collected data set (Supplementary Fig. 4e–g, see Methods section for details). For each system we estimated two different parameters: the time required to reach half of the final repressor level ($t_{1/2}$) and the delay between repressor up- or downregulation and effects on target gene expression ($\Delta t$). We assumed that the repressors were produced at a constant rate and degraded in a dTAG-13-dependent manner. For mCherry, the production rate was assumed to be a function of the repressor level and the associated parameters were estimated from the dTAG-13 titration experiments (Supplementary Fig. 3e). The mCherry degradation rate was assumed to be identical in all cell lines and was estimated from the

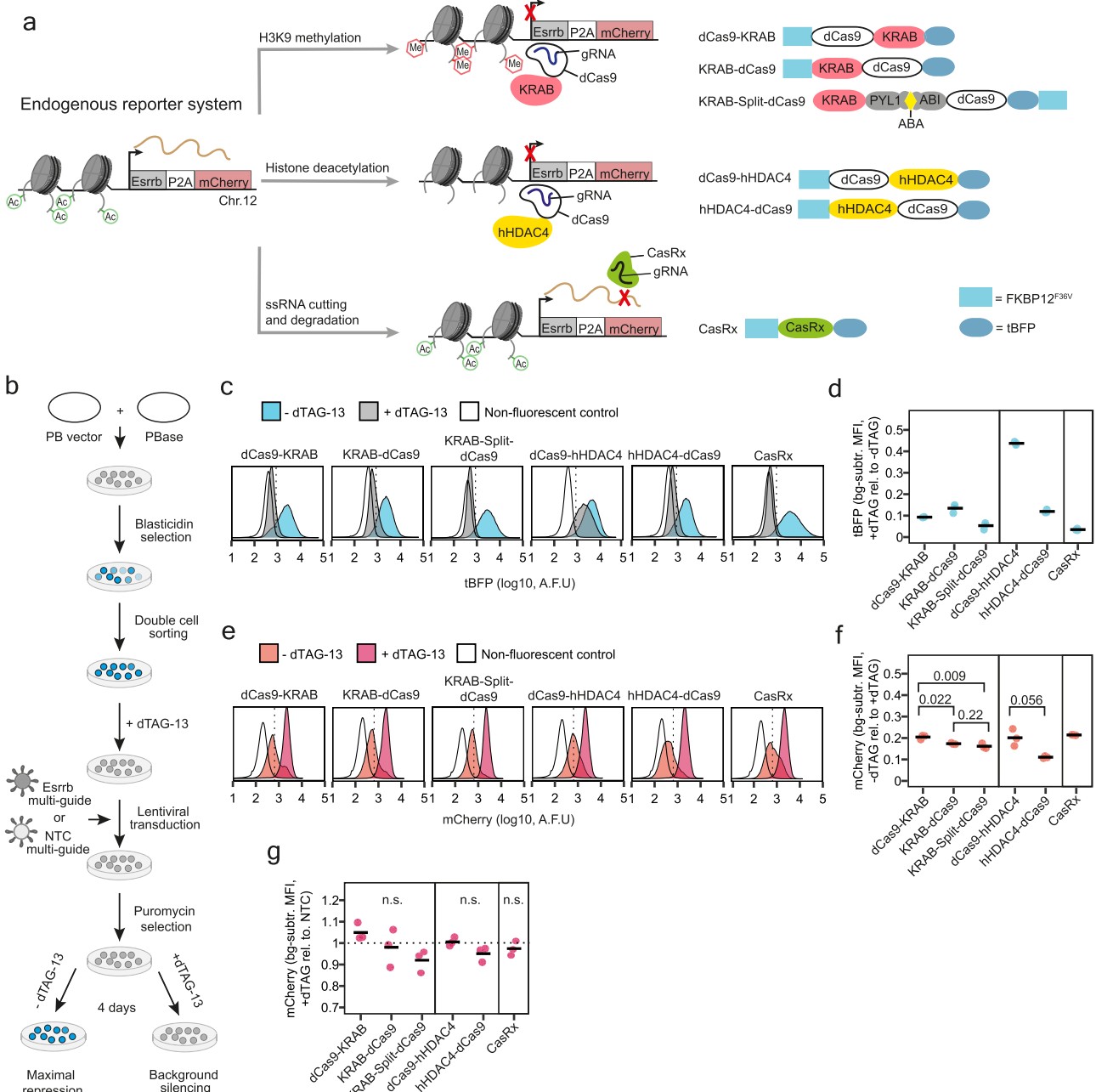

**Fig. 2 | Testing inducibility and efficiency of repression of degron-Cas-repressors with an endogenous reporter system. a** Three different repression mechanisms are compared with respect to their ability to repress the *Esrrb* gene, which is C-terminally fused with P2A-mCherry at its endogenous locus in the 1.8XX mESC line. For both KRAB- and hHDAC4-mediated repression, the repressor is tethered to the *Esrrb* promoter region via dCas9. KRAB-mediated repression induces H3K9 methylation (red hexagons), while hHDAC4 catalyses removal of histone acetylation (green circles). Both mechanisms repress transcription. CasRx binds *Esrrb* mRNA via complementary guide RNAs and cuts the transcript causing its degradation. The different designs compared in (**b**–**g**) are shown on the right. **b** Experimental strategy for testing inducibility and efficiency of repression of degron-Cas-repressor designs used in (**c**–**g**). Stable cell lines expressing the constructs shown in (**a**) together with Esrrb-targeting and non-targeting guides were generated as indicated. Cells are then either kept in medium with dTAG-13 (+dTAG-13, 500 nM) or without dTAG-13 (-dTAG-13, DMSO mock-treated) and analysed after 4 days by flow cytometry to assess background silencing and efficiency of

repression, respectively. In parallel to dTAG-13 removal, the KRAB-Split-dCas9 cell line is also treated for 4 days with 100 μM ABA to induce tethering of KRAB to dCas9. **c** Density plots of tBFP levels in cells containing Esrrb-targeting guides measured by flow cytometry. One biological replicate is shown. The dotted vertical line represents the 99th percentile of the non-fluorescent control. A.F.U. = arbitrary fluorescence units. **d** The degradation efficiency for each construct, quantified by comparing the tBFP levels in +dTAG-13 and -dTAG-13 conditions after subtracting the non-fluorescent control. **e** Same as in (**c**) but for mCherry. **f** Target repression quantified as the fold change in mCherry levels upon dTAG-13 withdrawal. *p* values of a two-sided *t*-test are reported. **g** Background repression quantified by comparing mCherry levels in cells with Esrrb-targeting and non-targeting guides in the presence of dTAG-13. No significant differences were observed between targeting and non-targeting guides (two-sided *t*-test). In (**d**, **f**, **g**) three biological replicates are shown as dots with a horizontal bar showing their mean. bg-subtr. = background-subtracted.

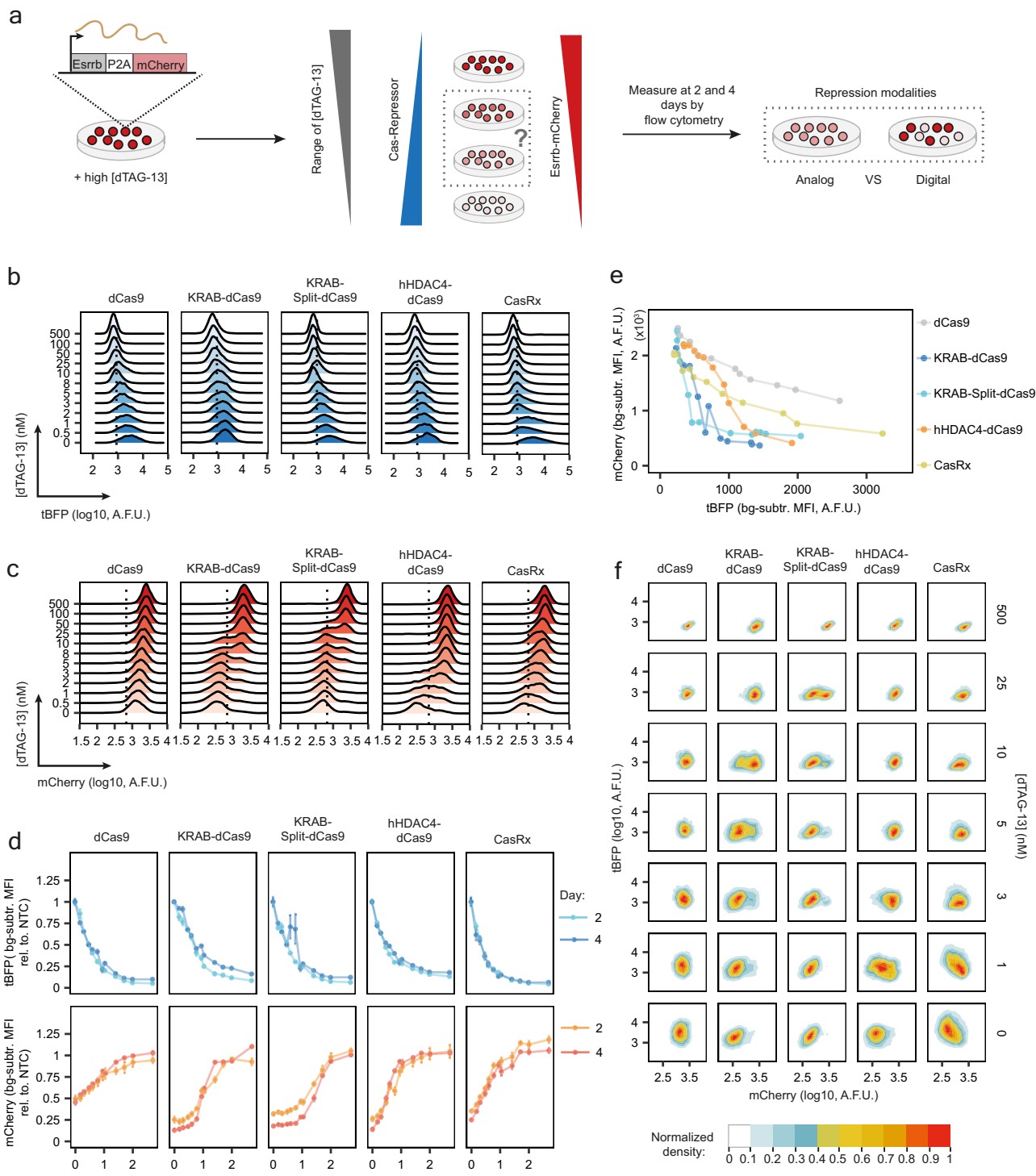

**Fig. 3 | Assessing tunability of degron-Cas-repressor systems. a** Experimental design: degron-Cas-repressor cell lines expressing ESRRB-mCherry are transferred from medium containing high concentration of dTAG-13 (repressor degraded) to media with a range of dTAG-13 concentrations. The degron-Cas-repressor levels are expected to decrease with increasing dTAG-13 concentrations, resulting in a rise in target gene expression (mCherry). Quantification of ESRRB-mCherry and degron-Cas-repressor (tBFP) levels after 2 and 4 days by flow cytometry will then allow the distinction between analog (homogenous intermediate levels) and digital (a mixture of positive and negative cells) repression. **b**, **c** Density plots of tBFP (**b**) and mCherry (**c**) expression levels measured by flow cytometry after 4 days of treatment. One biological replicate is shown. dCas9 and KRAB-Split-dCas9 are the same

construct (KRAB-Split-dCas9) with 100 μM ABA being added to the KRAB-Split-dCas9 4 days before the measurement. The dotted line shows the 99th percentile of non-fluorescent control cells. **d** tBFP (top) and mCherry (bottom) levels normalised to cells expressing NTC guides. The mean of three biological replicates (dots) ± standard deviation (vertical bars) is shown. bg-subtr. = background-subtracted; MFI = Median Fluorescence Intensity (**e**), mCherry MFI at different doses of repressor (tBFP MFI) after 4 days of titration. The mean of three biological replicates is shown. **f** Normalised 2D density plots showing tBFP and mCherry levels in populations of cells treated with different dTAG-13 concentrations (indicated on the right). A.F.U. = Arbitrary Fluorescence Units.

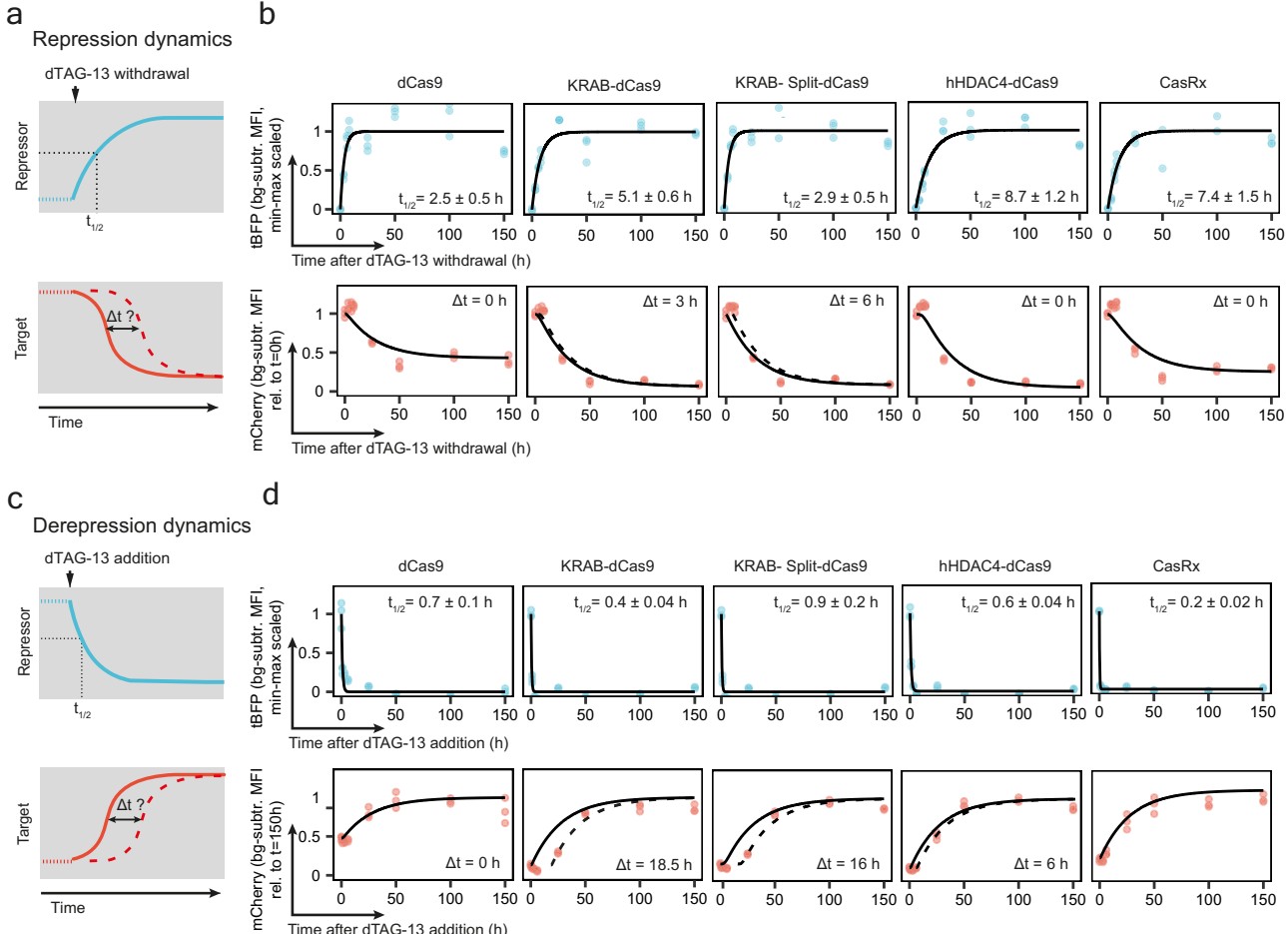

**Fig. 4 | Repression and derepression dynamics of degron-Cas-repressor systems. a** Schematics of the experimental setup to study the dynamics of repression for the different degron-Cas-repressors. dTAG-13 is withdrawn at time 0 and then repressor (tBFP) and target (ESRRB-mCherry) levels are measured by flow cytometry over time. Upon ligand withdrawal, the degron-Cas-repressor level increases to then reach a steady state. We estimate the time required to reach half of the maximal repressor level ($t_{1/2}$) by fitting an ordinary differential equation (ODE) to the experimental data. To assess whether target repression is immediate upon repressor upregulation ($\Delta t=0$, solid line) or occurs with a delay ($\Delta t>0$, dashed line), the mCherry data were fitted with an ODE model, where target gene expression varies as a function of time and of repressor concentration with or without assuming a delay between repressor upregulation and target gene repression ($\Delta t$). See Supplementary Fig. 4e and the methods section for details on the modelling approach. **b** Bg-subtr. MFI for repressor (tBFP, scaled between 0 and 1) and target gene (mCherry, rel. to initial time point) for the different degron-Cas-repressor systems, for three biological replicates (dots). The black lines in each plot represent the best fit of the ODE model. For tBFP, the $t_{1/2} \pm$ its standard error is indicated for each degron-repressor system. For mCherry, the continuous line shows the result of the ODE model with $\Delta t = 0$ h and the dashed line the ODE model with the $\Delta t$ that minimises the Mean Absolute Error of the model compared to the experimental measurements. **c** To study the dynamics of derepression associated with each system, dTAG-13 is added back to the medium after 4 days in the absence of dTAG-13 (in the presence of repression). The same type of model as shown in (**a**) is used and we estimate the delay between repressor degradation and target gene derepression. **d** Same as in (**b**) but for the derepression dynamics experiment schematised in (**c**). The two KRAB-based repression systems show a substantial delay in target gene derepression.

derepression time course for CasRx under the assumption that release of repression was immediate in this system.

The time required to reach half of the maximal repressor level upon ligand withdrawal ($t_{1/2}$) ranged from ~3 h for dCas9 and KRAB-Split-dCas9 to ~7–8 h for CasRx and hHDAC4-dCas9 (Fig. 4b top). These differences are likely due to variation in protein turnover rates in the absence of ligand, because those determine the dynamics of upregulation. Once the repressor was upregulated, we observed immediate target gene repression when using dCas9, hHDAC4-dCas9 and CasRx ($\Delta t = 0$ h), while a small delay was detected for the two KRAB-mediated systems ($\Delta t = 3$–6 h, Fig. 4b bottom). Hence, our analyses show that the speed of target depletion is mainly determined by the repressor dynamics (and target stability) and that full repression is established within 2 days for all systems.

When analysing derepression upon dTAG-13 addition after 4 days of culture without the degrader, all constructs were rapidly depleted

with a half time of <1 h (Fig. 4d, top). Although repression was fully reversible in all cases, we observed clear differences in the dynamics. While derepression was immediate for dCas9 alone, a delay of 16–18 h was observed for the KRAB systems, which was substantially smaller (6 h) for hHDAC4-dCas9 (Fig. 4d, bottom). In summary, we have estimated the dynamics of repression and derepression in our reporter system for different degron-Cas-repressors. While target repression appears to be immediate, derepression dynamics depend on the mode of repression.

**CasTuner can tune endogenous gene expression in human cells**
Having demonstrated tunability of CasTuner in mESCs, we decided to test the system in more differentiated human cells. We employed a previously generated HeLa cell line, where the *STAG2* gene (subunit of the Cohesin complex) is endogenously tagged with EGFP at its C-terminus[38]. We generated cell lines, where *STAG2* was targeted by the

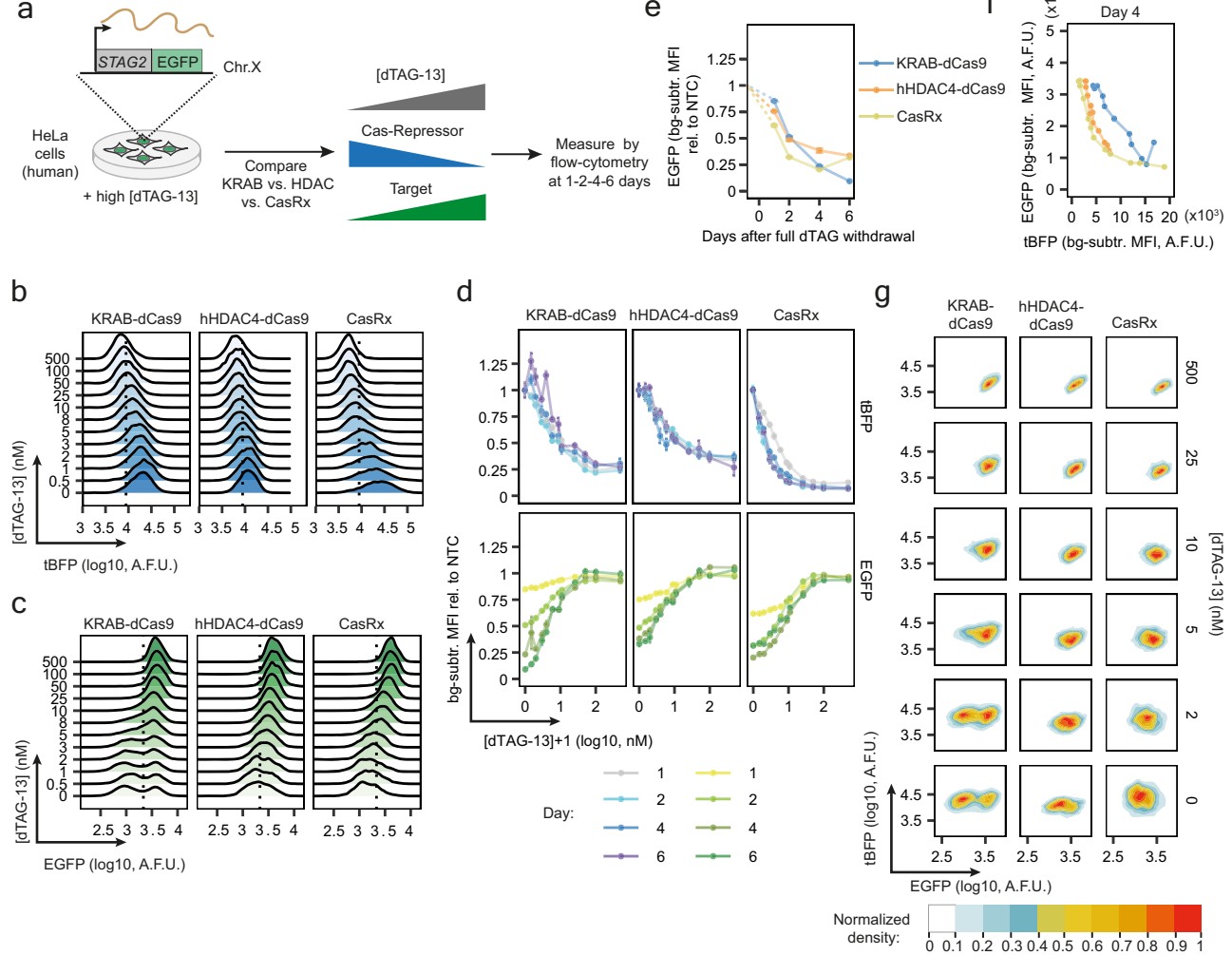

**Fig. 5 | CasTuner can tune endogenous gene expression in human cells.**
**a** Experimental scheme: STAG2-EGFP HeLa Kyoto cells stably expressing either (degron-)KRAB-dCas9, hHDAC4-dCas9 or CasRx together with the respective guide RNAs targeting STAG2 were grown in high dTAG-13 (500 nM) media and then titrated with a range of dTAG-13 concentrations. The levels of STAG2-EGFP and degron-Cas-repressor (tBFP) are measured after 1, 2, 4 and 6 days. **b, c** Density plots of tBFP (**b**) and EGFP (**c**) expression levels measured by flow cytometry, in cells expressing *STAG2*-targeting guides after 4 days of treatment. One biological replicate is shown. **d** tBFP (top) and EGFP (bottom) at different dTAG-13 concentrations at the indicated time points (colours). The mean of three biological replicates (dots) ± standard deviation (vertical bars) is shown. **e** Comparison of target gene (EGFP) repression dynamics upon complete dTAG-13 withdrawal. **f** Comparison of repressor potency by showing target gene expression (EGFP) vs repressor abundance (tBFP) at the day 4 time point. The mean of three biological replicates is shown. **g** Normalised 2D density plots showing tBFP and EGFP levels in populations of cells treated with different dTAG-13 concentrations (indicated on the right). A.F.U. = Arbitrary Fluorescence Units.

two CasTuner systems or FKBP12$^{F36V}$-KRAB-dCas9 for comparison. We treated the cells with a range of dTAG-13 concentrations and measured the effects on repressor abundance (tBFP) and STAG2-EGFP by flow cytometry at different time points (Fig. 5a–d, Supplementary Fig. 5a,b). All systems could be homogeneously titrated (tBFP), but again striking differences in the distribution of target gene repression (EGFP) were observed (Fig. 5b,c). While KRAB-dCas9 induced a bimodal distribution, the CasTuner systems repressed the target gene in a more homogeneous manner (Fig. 5c, Supplementary Fig. 5c, Hall-York test for unimodality, $p < 0.01$, see methods for details). These differences would not be visible in bulk measurements (Fig. 5d).

When analysing repression dynamics, CasRx acted the fastest with near full repression at day 2, while KRAB-dCas9, where repression kept increasing until day 6, was the slowest (Fig. 5e). In particular the repression dynamics of the KRAB system appeared to be significantly slower in HeLa cells compared to mESCs, where full repression was already reached after 48 h (Fig. 4b). The fact that CasRx and dCas9 use different guide RNAs makes a direct comparison of their potency

difficult, but the KRAB and the hHDAC4 systems can be compared directly. Among the two, KRAB-dCas9 ultimately led to the strongest repression (Fig. 5e), possibly because the system was expressed at higher levels. In fact, when comparing repressor abundance (BFP) and repression strength (EGFP), hHDAC4-dCas9 appeared to be more potent than KRAB-dCas9 (Fig. 5f, showing a comparison at day 4). Two-dimensional visualisation of repressors and target gene levels showed similar tBFP expression in the EGFP-positive and negative cells for KRAB-dCas9, again suggesting that KRAB-mediated silencing is a probabilistic process (Fig. 5g). Collectively, these results show that CasTuner can be used to homogeneously tune endogenous gene expression also in more differentiated human cell lines.

## CasTuner can quantify dose-dependent effects of NANOG and OCT4 on target genes and cellular phenotypes

We designed CasTuner in order to finely regulate the quantity of a gene product. This allows us to study how gene dose relates to a phenotype of interest (dose-response). As a proof of concept, we applied

CasTuner to titrate the dose of two core pluripotency factors, NANOG and OCT4 (POU5F1), in mESCs and measured effects on target gene expression and cellular phenotype. For both factors threshold levels have been suggested to exist that determine whether cells maintain pluripotency or undergo differentiation[7,8,39–41].

We employed a mESC line (1.8XX-Nanog-mCherry) where the endogenous *Nanog* gene is homozygously tagged at its C-terminus with P2A-mCherry[36]. We stably integrated the FKBP12$^{F36V}$-hHDAC4-dCas9 system as described above and transduced the cells in the absence of CasTuner expression (high dTAG-13) with lentiviral multi-guide plasmids targeting the *Nanog* or *Oct4* promoters.

To titrate NANOG levels, we first tested a range of dTAG-13 concentrations and analysis time points (Supplementary Fig. 6a–d). For 5 selected concentrations we analysed target gene expression and the capacity of self-renewal after 3 days of dTAG-13 titration (Fig. 6). We confirmed that NANOG was indeed titrated both at the mRNA and protein level (Fig. 6b,c). We then used qRT-PCR to quantify the expression of NANOG target genes *Esrrb*, *Rex1* (*Zfp42*), *NrbO1*, *Sox2* and *Klf4*, which have been reported to be activated by NANOG[42], and of *Xist*, which is known to be repressed by NANOG[43] (Fig. 6d and Supplementary Fig. 6e). To assess the phenotypic consequences of NANOG titration, we seeded cells at clonal density and stained the colonies 6 days later for alkaline phosphatase (AP) activity as a marker for the undifferentiated state (Fig. 6e,f, Supplementary Fig. 6f). We then counted the total number of colonies (Fig. 6e) and also categorised them as undifferentiated, differentiated or mixed (Fig. 6f).

To quantify the dose-response relationships, we fitted a Hill curve for each target gene and for the phenotypic readout (% undifferentiated cells). In this way, we estimated two parameters: (1) the dose sensitivity $[N]_{1/2}$, given by the NANOG reduction required to induce half of the maximal effect, and (2) the degree of non-linearity of the response given by the Hill coefficient n, where a high value (n»1) indicates a switch-like response, and a lower value a more gradual mode of regulation (see Methods). We observed marked differences in the dose-response curves between the tested targets (Fig. 6d). *Esrrb* and *Rex1* were the most linearly related to NANOG dose (smallest n). *NrOb1* reacted most sensitively with $[N]_{1/2} = 0.55$, while *Xist* required a much stronger NANOG reduction to be affected ($[N]_{1/2} = 0.21$), with the other tested targets lying in between these two cases. The cellular phenotype exhibited a dose-dependency similar to the majority of targets with a clear loss of clonogenicity and induction of differentiation, when Nanog levels dropped below ~30% (Fig. 6e+g). We can conclude that the sensitivity to NANOG dose is variable among target genes. Some targets thus respond already, when no phenotypic consequences are detected, but overall we see a good agreement between target gene expression and loss of self-renewal capacity.

For the second factor we titrated, OCT4, a series of previous studies have shown that mESCs respond sensitively to dosage alterations. While overexpression drives primitive endoderm differentiation, a small reduction in heterozygous mutant cells enhances the pluripotent state and stronger depletion induces dedifferentiation towards trophectoderm (TE)[7,8,41]. We titrated Oct4 using 5 dTAG-13 concentrations (Fig. 7a). We indeed observed induction of the TE program and loss of mESC markers, when Oct4 levels dropped below ~30% (Fig. 7b, Supplementary Fig. 7a). Notably, the mESC marker Nr0b1 responded most sensitively ($[O]_{1/2} = 0.68$), while Sox2 was again very robust to dosage alterations ($[O]_{1/2} = 0.29$), mirroring our observation upon NANOG titration (Fig. 7c vs Fig. 6d). Among the tested marker genes, Nanog was the only one that showed a non-monotonic dose-response curve (Fig. 7c and Supplementary Fig. 7b,c). Nanog levels increased, when Oct4 was reduced by ~50% and decreased again at lower levels. This observation is in agreement with a previously reported increase in Nanog levels in heterozygous Oct4 mutant mESCs[8,41] and was accompanied by an enhanced naive-like morphology at intermediate Oct4 levels (Supplementary Fig. 7d).

Taken together, our results show that CasTuner can be used to uncover dose-dependent effects on target genes and cellular phenotypes. Since its application is much easier than genetic perturbations, which have previously been used to investigate Oct4, our system will facilitate the study of dose-dependent effects for a larger number of factors in the future.

## Discussion

In this study we have developed a toolkit, CasTuner, based on Cas-derived repressors fused with a conditional degron domain. CasTuner allows quantitative control of endogenous gene expression either at the transcriptional (with hHDAC4-dCas9) or post-transcriptional level (with CasRx). The system employs the FKBP12$^{F36V}$ degron to precisely control Cas-repressor abundance by varying the concentration of the dTAG-13 ligand. We show that intermediate CasTuner levels can homogeneously tune target gene expression in an analog manner, as opposed to the widely used KRAB-mediated CRISPRi system, which exhibits a digital mode of action. Analysis of repression and derepression dynamics suggests that CasTuner might generally act more rapidly compared to the KRAB system with the differences being more pronounced in HeLa cells compared to mESCs. Since CasTuner can titrate endogenous gene activity in a physiologically relevant range, the toolkit will allow us to gain a better understanding of how cellular functions are quantitatively controlled in mammals.

As a proof-of-concept, we employed CasTuner to titrate the core pluripotency factors NANOG and OCT4 in mESCs and measured dose-response curves for their target genes, lineage markers and the cellular phenotype. We show that sensitivity and also non-linearity of the response is variable between targets. Different scenarios could be envisioned to explain these differences. NANOG and OCT4 might bind their target genes with varying affinity, such that a subset would remain occupied even at lower concentrations. This could be due to differences in sequence composition or spatial arrangements of binding sites. Alternatively, NANOG and OCT4 might co-bind with other factors, which could lead to differential sensitivity depending on which cofactor is employed. For Xist, where NANOG binds together with multiple other factors including SOX2[43,44], repression might for example only be released, once NANOG and SOX2 are depleted, which only occurs at lower NANOG levels.

An unusual non-monotonic dose-response relationship was found between OCT4 and NANOG, where NANOG was upregulated at intermediate OCT4 levels but reduced at lower levels. This finding is in agreement with previous reports analysing heterozygous Oct4 mutant cell lines[8,41]. The observed NANOG upregulation is seemingly at odds with the fact that OCT4 binds together with SOX2 to the *Nanog* promoter, thereby promoting its expression[41,45]. However, it has also been shown that cells with intermediate OCT4 levels upregulate components of the Wnt pathway such as Wnt3a, which in turn upregulate NANOG[41]. The balance between loss of direct activation and gain of indirect activation might thus determine whether a given OCT4 dose will result in up- or downregulation of NANOG. Quantitative tuning of transcription factor abundance can thus reveal complex regulatory relationships within gene networks.

Our results reveal that certain repression mechanisms, such as histone deacetylation and RNA degradation function in an analog way in the sense that they can induce stable intermediate expression levels when analysed with single-cell resolution. Others, such as KRAB-mediated systems, work in a digital manner, where the quantity of the repressor only defines the percentage of cells that will shut off target gene expression. In the latter case, bimodal distributions are observed for the targeted gene at intermediate repressor levels. Such bimodal patterns are typical for bistable systems, which usually respond in a switch-like manner[46]. In the context of gene regulation, bistability can arise from positive feedback loops in chromatin regulation[47]. Here histone-modifying enzymes are recruited through binding to the

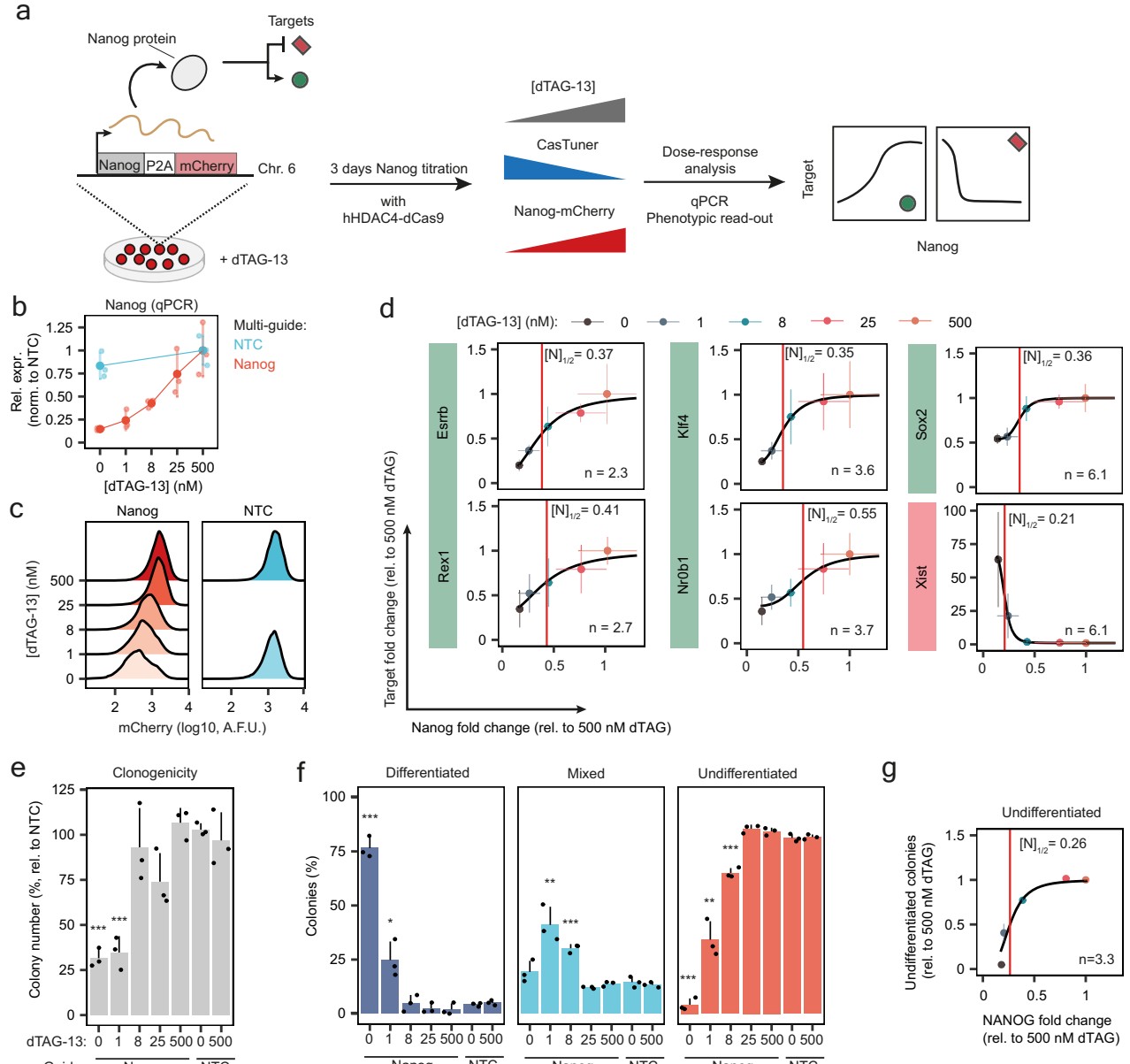

**Fig. 6 | NANOG dose-response curves measured using CasTuner. a** Experimental strategy to measure dose-response curves between NANOG and its target genes. A mESC line with *Nanog* fused with P2A-mCherry at its endogenous locus, stably expressing CasTuner (FKBP12^F36V-hHDAC4-dCas9), was transduced with Nanog-targeting or non-targeting guides in the presence of dTAG-13. After a 3-day treatment with a range of dTAG-13 concentrations, NANOG target genes and cellular phenotype were assessed. **b** Nanog expression normalised to cells with NTC guides at 500 nM dTAG-13, measured by qRT-PCR, for 3 biological replicates (small dots); the mean (big dots) and s.d. are shown. **c** mCherry levels as a proxy for NANOG protein measured by flow cytometry. Only one replicate was measured and the experiment was performed in the exact same conditions as in (**b**). A.F.U. = arbitrary fluorescence units. **d** Dose-response curves for NANOG target genes. The mean and s.d. of 3 biological replicates together with the best fit with a Hill function (black line) is shown. The red vertical line marks the fold change of *Nanog* that leads to half of the maximal expected fold change variation of the target ([N]$_{1/2}$), according to the

fitted model, which is also reported in each plot, together with the Hill coefficient n. The $R^2$ for the fits ranges between 0.94 and 0.99. **e**, **f** After a 3-day treatment with the indicated dTAG-13 concentrations cells were seeded at clonal density and stained for Alkaline Phosphatase (AP) activity 6 days later. The number of colonies (**e**) and their categorisation in differentiated, mixed or undifferentiated clones based on the AP staining (**f**) is reported. The mean and s.d. of three biological replicates is shown. Black dots denote the individual measurements. A two-sided *t*-test between each dTAG-13 condition compared to the 500 nM dTAG-13 condition was performed. *** = *p* value < 0.001; ** = *p* value < 0.01; * = *p* value < 0.05. **g** The percentage of undifferentiated colonies (from **f**) normalised to the mean percentage in the 500 nM dTAG-13 condition, plotted against the average NANOG fold change measured by flow cytometry before seeding the cells at clonal density. Mean and s.d. are shown. A Hill equation (black curve) was fitted and the parameters reported similarly to (**d**).

---

modification they deposit either directly or in an indirect manner through additional proteins. Such feedbacks have been found in particular for repressive modifications, such as H3K27me3, which is deposited by the PRC2 complex[48], and H3K9me3, which is catalysed by several enzymes in mammalian cells, including SETDB1, which is recruited by the KRAB repressor domain via a KAP1-mediated

interaction[49]. Such a mechanism is also thought to allow spreading of the repressed state along the DNA to form chromatin domains, and to underlie epigenetic memory[34,47,50]. Although silencing is readily reversible in mESCs for all repressors we tested, the observed slower derepression for the KRAB systems might be due to bistability, which seems to confer (short-term) epigenetic memory. Notably,

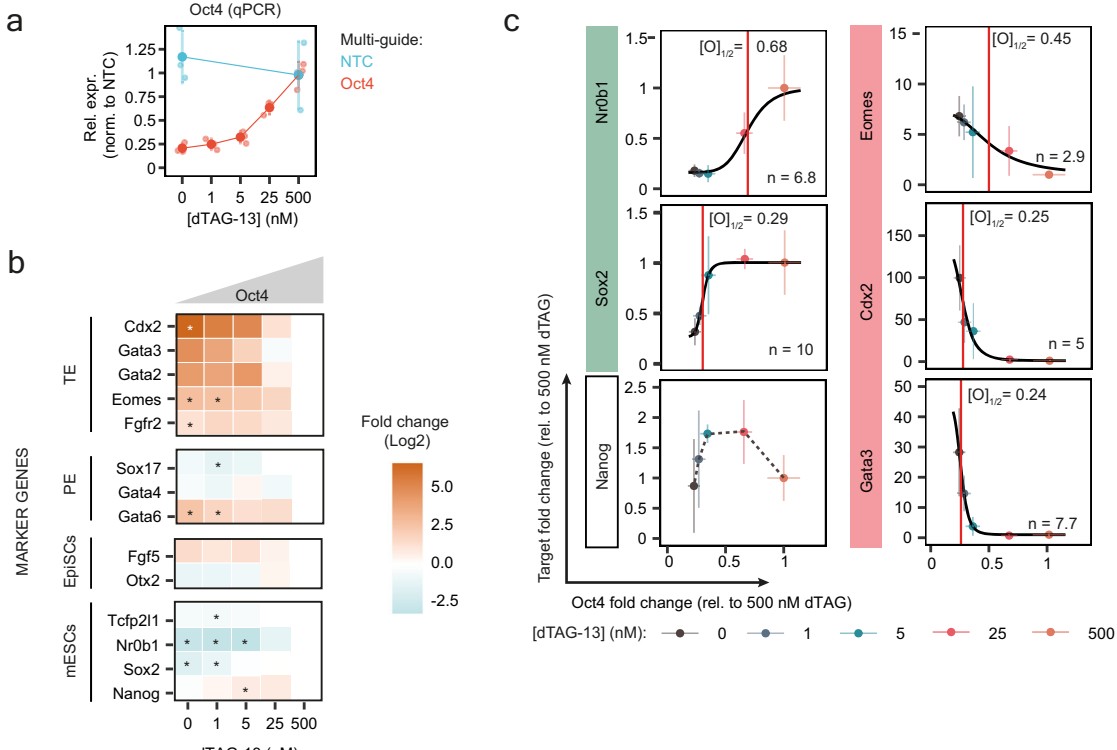

**Fig. 7 | Titrating Oct4 with CasTuner reveals a threshold level to induce the trophectoderm transcriptional program. a–c** Oct4 titration in mESCs (1.8XX Nanog-mCherry cell line expressing FKBP12^F36V^-hHDAC4-dCas9) for 3 days. **a** Oct4 expression fold change at different dTAG-13 concentrations as measured by qRT-PCR, for 3 biological replicates (small dots). Vertical lines indicate the standard deviation and larger dots the mean. **b** Heatmap of average expression fold change at different dTAG-13 concentrations relative to the 500 nM dTAG-13 condition for the indicated lineage markers. mESCs = mouse embryonic stem cells; EpiSCs = epiblast-like stem cells; PE = primitive endoderm; TE = trophectoderm. A two-sided *t*-test between each dTAG-13 condition to the 500 nM dTAG-13 condition was used. Asterisks indicate *p* value < 0.05. Exact *p* values are given in Source Data. **c** Dose-

response relationship between *Oct4* and target gene levels measured by qRT-PCR, shown as the mean of three biological replicates (dots). Horizontal and vertical lines indicate the standard deviation for Oct4 and its target genes, respectively. The black curve represents the most likely Hill curve fitted using a non-linear least square approach (see Methods). For all shown target genes, except Nanog, the red vertical line marks the fold change of Oct4 that leads to half of the maximal expected fold change variation of the target ([O]$_{1/2}$), according to the fitted model, which is also reported in each plot, together with the Hill coefficient n. The $R^2$ for the fits ranges between 0.92 and 0.99. The Oct4-Nanog dose-response relationship is non-monotonic and a dashed black line is used to connect the mean of the different average measurements.

derepression dynamics appear to be substantially slower in other more differentiated cell types[26,51,52]. Similarly, our comparison in HeLa cells suggested that repression is induced significantly slower in this cell type, when employing the KRAB domain compared to CasTuner. While KRAB systems induce a repressed chromatin state, hHDAC4 in the CasTuner toolkit functions via removal of active histone modifications, namely acetylation of a multitude of histone residues[53]. Erasure of the active chromatin state at target gene promoters thus appears to enable rapid analog tuning as opposed to induced heterochromatinisation.

The CasTuner toolkit also provides a chromatin-independent approach to tune expression levels, where RNA degradation is manipulated through the CasRx system. This approach is similar to direct degron-tagging of a target gene, since both modulate protein abundance without affecting transcription. Indeed, direct degron-tagging would be an alternative approach to homogeneously titrate protein levels, which would allow faster response dynamics compared to CasTuner, where the response time is limited by the repressor half-life, in addition to the mRNA and protein half-life of the target gene. Such an approach has recently been used to assess dose-dependent effects of SOX9[54]. Direct degron-tagging is however associated with some major drawbacks, such as the often-observed basal (uninduced) degradation[55], possible impediments to structural folding and interactions and labour-intensive generation of the required cell lines through gene targeting. With CasTuner, titration of a new target gene requires only a single guide-encoding plasmid. The approach is

therefore relatively simple and cost-effective, allowing the study of how the quantity of different genes relates to a phenotype, thus preserving the scalability that characterise CRISPR-based technologies.

Of note, some studies recently reported that CasRx possesses collateral activity (i.e. unspecific cleavage of RNAs nearby the target RNA), particularly when targeting highly expressed transcripts or transcripts with tandem repeats[56,57]. It has been shown that reducing the level of CasRx itself can mitigate these effects[56]. The chemical control of CasRx levels in our system could thus help to tame unwanted collateral effects in more challenging applications. Alternatively, engineered variants of Cas13 with reduced collateral activity have been recently reported, which could be easily used to substitute the original CasRx used in our study[58].

In the future, CasTuner could also be expanded to an orthogonal system. Since we identified two potent degron domains (AID and FKBP12^F36V^) and two tunable repressor systems (dCas9-hHDAC4 and CasRx), which employ different types of guide RNAs, each repressor could be controlled by a different degron domain. In this way, treatment with variable concentrations of different degrader molecules would allow simultaneous independent tuning of two endogenous genes. An interesting application for CasTuner and also for a CasTuner-based orthogonal system could be the quantitative analysis of phase separation phenomena in cells. Although phase separation has been suggested as regulatory mechanisms for a variety of cellular processes, its occurrence and functional importance in vivo is often debated[59].

A hallmark of phase separation is the existence of a saturation concentration for the involved macromolecules, above which phase separation occurs[60]. Testing the existence of a saturation concentration in cells requires the ability to titrate macromolecules such as proteins or RNAs in vivo. Here CasTuner would be a powerful tool to address this technical challenge in the growing phase separation field.

## Methods

### Cell lines

The female TX1072 mESC line is a F1 hybrid ESC line derived from a cross between the 57BL/6 (B6) and CAST/EiJ (Cast) mouse strains that carries a doxycycline-responsive promoter in front of the *Xist* gene on the B6 chromosome and an rtTA insertion in the *Rosa26* locus[61]. The 1.8XX *Nanog*-mCherry and *Esrrb*-mCherry reporter lines are female mESC lines that carry a homozygous insertion of 7xMS2 repeats in *Xist* exon 7 and a C-terminal P2A-mCherry tag at the *Nanog* or *Esrrb* genes, respectively[36]. In the STAG2-EGFP HeLa Kyoto cell line (a gift from Jan-Michael Peters) the *STAG2* gene is homozygously tagged with EGFP at the C-terminus of the endogenous locus[38,62].

All Cas-Repressor cell lines were generated through piggyBac transposition, antibiotic selection with blasticidin and FACS based on tBFP fluorescence levels (see below).

### mESCs culture

All mESC lines were grown without feeder cells on gelatin-coated flasks (Millipore, 0.1%). mESCs were passaged every second day at a density of $4 \times 10^4$ cells/cm$^2$ and medium was changed daily. Cells were grown in serum-containing medium (DMEM (Sigma), 15% ESC-grade FBS (Gibco), 0.1 mM β-mercaptoethanol), supplemented with 1000 U/ml leukaemia inhibitory factor (LIF, Millipore) only for all experiments performed in 1.8XX mESCs or supplemented with LIF and 2i (3 μM Gsk3 inhibitor CT-99021, 1 μM MEK inhibitor PD0325901, Axon), when growing TX1072 cell lines. For the NANOG and OCT4 titration experiments, the cells were seeded at a lower density ($3 \times 10^4$ cells/cm$^2$) and not passaged before RNA harvesting, to counteract possible selective effects of *Nanog* or *Oct4* knock-down. For experiments with a flow cytometry readout, cell treatment and analysis was usually performed in 96-well plates. Here cells were seeded at a density of 20,000 cells per well and were passaged 1:8 after 2 days for longer treatments.

### HeLa cell culture

HeLa cells were grown in DMEM (Gibco), 10%FBS (Gibco), 100U/ml penicillin, 100 μg/ml streptomycin, 10 mM HEPES (Gibco). Cells were passaged every 2–3 days at a 1:5 or 1:6 dilution, respectively. The flow-cytometry experiments were performed in 96-well plates seeding 10,000 cells per well and passaged 1:5 after 2 days for longer treatments.

### Alkaline phosphatase assay

For the NANOG titration experiment, cells were first seeded at a density of $3 \times 10^4$ cells/cm$^2$ for 3 days at different dTAG-13 concentrations. The cells were then seeded at clonal density (500–1000 cells per well in a 6-well plate) and grown for 6 days, maintaining the same ligand concentrations and changing the medium every day. After 6 days, colonies were fixed for 30 s with Citrate-Acetone-Formaldehyde solution (25% Citrate solution (Sigma, #854), 67% Acetone (Merck, #67-64-1), 8% Formaldehyde (Sigma, #F8775)) and stained for 15 min with Alkaline Phosphatase staining kit (Sigma, #86R-1kt). The total number of colonies and the number of undifferentiated, mixed and differentiated colonies were counted using a stereomicroscope. Raw counts are given in Supplementary Data 2.

### PiggyBac transposition

In order to generate cell lines stably expressing dCas9 and CasRx constructs, expression plasmids were genomically integrated through piggyBac transposition. To this end mESCs were transfected using Lipofectamine™ 3000 Transfection Reagent (Invitrogen) with the donor plasmid and a plasmid encoding for a hyperactive piggyBac transposase (pBROAD3-hyPBase-IRES-zeocin, a kind gift from the Giorgetti lab) in a 5:1 molar ratio using a total of 2.5 μg of DNA. A reverse transfection protocol was employed, where $0.4 \times 10^6$ cells are seeded together with the lipofection mixture containing the plasmids in a 6-well plate coated with gelatin. The lipofection mixture was prepared according to the manufacturer's instructions. On the next day, fresh medium was added to the cells. On the second day cells were transferred to a T25 flask with medium containing blasticidin (5 ng/μl, Roth), followed by selection for 7 days. During antibiotic selection, the medium was changed daily and cells were passaged when they became ~80% confluent, to favour an even selection of all cells. HeLa cell lines carrying dCas9 and CasRx constructs were generated in the same way, except that they were seeded at $0.2 \times 10^6$ cells in 6-well plates for reverse lipofection.

### FACS

For Fluorescence Associated Cell Sorting (FACS), a BD FACSAria Fusion sorter (Beckton Dickinson, IC-Nr.:68198, Serial-Nr.:R658282830001) with a 2B-5YG-3R-2UV-6V lasers configuration was used. The cells were sorted based on their tBFP level. For an example of the strategy employed to sort cells for the testing of different degrons fused to dCas9, see Fig. 1A. An example of the gating coordinates used for double sorting of cells with degron-Cas-repressor systems is given in Supplementary Data 3. The strategy for sorting was selected in order to obtain a high number of cells, with an as uniform as possible level of expression, while maintaining high expression levels of the construct, clearly distinguishable from the fluorescent background of the cells. For mESCs, which we normally do not grow in the presence of antibiotics, immediately after sorting, cells were centrifuged, resuspended in the appropriate medium with 1× Penicillin-Streptomycin (Gibco, #15070063) and seeded. Penicillin-Streptomycin was kept for 2 passages.

### Lentiviral transduction

For the generation of cell lines carrying CRISPRi or CasRx multi-guide plasmids, we used lentiviral transduction. DNA constructs were first packaged into lentiviral particles. For this, $1 \times 10^6$ HEK293T cells were seeded into one well of a 6-well plate and transfected the next day with the lentiviral packaging vectors: 1.2 μg pLP1, 0.6 μg pLP2, and 0.4 μg VSVG (Thermo Fisher Scientific), together with 2 μg of the desired construct using Lipofectamine 2000 (Thermo Fisher Scientific). HEK293T supernatant containing the viral particles was harvested after 48 h. The viral supernatant was concentrated using Lenti-X™ Concentrator (#631232, Takara) according to the manufacturer's instructions, resuspended in 200 μl of mESC or HeLa medium and frozen at −80 °C. $0.2 \times 10^6$ cells were seeded per 12-well (mESCs) or 6-well (HeLa cells) in medium containing 500 nM dTAG-13 (Tocris) and transduced the next day with 50 μl of concentrated viral supernatant and 8 ng/μl polybrene (Sigma). Antibiotic selection with 1 μg/ml puromycin (Sigma) was started 2 days after transduction and kept until all cells in the non-transduced control were dead (typically within 1–2 passages).

### Ligands

Auxin (3-Indoleacetic acid, IAA, GoldBio #I-110-25) was dissolved in EtOH to a 400 mM stock dilution, according to the manufacturer's instructions. Aliquots were kept at −20 °C, protected from light. dTAG-13 (Tocris #6605) was dissolved in DMSO to a 5 mM stock concentration, according to manufacturer's instructions. Aliquots were kept at −20 °C. Trimethoprim (TMP, Sigma #T7883) was dissolved in DMSO to a concentration of 200 mM and stock aliquots were kept at −20 °C. Asunaprevir (ASN, MedChem Express #HY-14434) was dissolved in DMSO to 10 mM stock solution and stored at −20 °C.

(Z)−4-Hydroxytamoxifen (4OHT, Sigma #H7904) was dissolved in EtOH to a stock solution of 10 mM and stored at 4 °C.

### sgRNAs design

sgRNAs to target the *Esrrb*, *Nanog* and *STAG2* promoters were designed using the CRISPR-Cas9 online tool Chopchop (https://chopchop.cbu.uib.no/)[63]. Because *Esrrb* has multiple isoforms with different transcription start sites, the mESC-specific isoform (ENSMUST00000115313.7) was used in the query. Four of the highest ranking guides were selected, avoiding to pick guides in close proximity between each other (<50 base pairs) and/or containing BsmBI restriction sites. *Oct4/PouSf1* guides were designed based on the results of a currently ongoing pooled CRISPR screen. Non-targeting control (NTC) sgRNAs were extracted from previous publications[64] and are predicted to target non-functional genomic regions.

For CasRx-mediated repression, the online tool https://cas13design.nygenome.org/ was used to design guides[65]. For Esrrb, the shortest translated transcript isoform that contains all constitutive exons was used in the query (ENSMUST00000167891.1) and the 3 highest ranking guide sequences were selected. For STAG2, we targeted the isoform with highest expression in HeLa cells, according to RNA-seq data provided by the same tool. As negative controls we extracted safe-targeting control guide sequences from a previous publication and removed 4 nucleotides at the 5′-end[66], to obtain 23 base pairs long sequences. The resulting NTC sequences were confirmed to have low similarity with sequences contained in the NCBI Transcript Reference Sequences database for mus musculus, using Blast https://blast.ncbi.nlm.nih.gov/Blast.cgi. The guide sequences used in this study are provided in Supplementary Data 1 (part A for dCas9 and part B for CasRx).

### Cloning of sgRNAs in multi-guide expression vectors

Multi-guide plasmids for CRISPRi were cloned as previously described[36]. Briefly, four different sgRNAs were cloned into the sgRNA expression plasmid SP199[36], with Golden Gate cloning, such that each sgRNA is controlled by a different Pol III promoter (hU6, mU6, 7SK and hH1).

### Cloning of CasRx guide arrays

CasRx guides were cloned as an array into the pLentiRNAGuide_001 - hU6-RfxCas13d-DR1-BsmBI-EFS-Puro-WPRE plasmid, which was a gift from Neville Sanjana (Addgene plasmid # 138150; http://n2t.net/addgene:138150; RRID:Addgene_138150). To this end, we designed a sequence containing three 23 base pair-long guides interspaced by 2 optimised direct repeats, previously described[66] and BsmBI restriction sites to produce compatible overhangs to the entry vector. Such sequence was produced by PCR amplifying a 120 bp-long oligonucleotide (template) using two compatible primers to extend the sequence to its final length (157 bp, including 3 additional nucleotides at each site to favour restriction enzyme activity). Insert and pLentiRNAGuide_001 were linearised with the BsmBI restriction enzyme and ligated using T4 DNA ligase (NEB #M0202S). Primers and templates are provided in Supplementary Data 1, part C.

### Cloning of PiggyBac plasmids

All piggyBac plasmids used in this study have been obtained with standard molecular cloning techniques using as backbone pSLQ2812 (addgene #84240, a kind gift from the Qi lab). The ecDHFR degron domain was amplified from CAG-DDdCas9VP192-T2A-EGFP-ires-puro (Addgene plasmid # 69534; RRID:Addgene_69534, a gift from Timo Otonkoski). The SMASh degron domain was amplified from pCS6-SMASh-YFP, which was a gift from Michael Lin (Addgene plasmid # 68852; RRID:Addgene_68852). The ER50 degron domain was amplified from pBMN ER50-YFP, a gift from Thomas Wandless (Addgene plasmid # 37259; RRID:Addgene_37259). The AID degron domain was amplified from pcDNA5-H2B-AID-EYFP, a gift from Don Cleveland (Addgene plasmid # 47329; RRID:Addgene_47329). The OsTIR1 protein employed for the AID and mAID degron systems was amplified from pMK232 (CMV-OsTIR1-PURO), which was a gift from Masato Kanemaki (Addgene plasmid # 72834; RRID:Addgene_72834). The KRAB-Split-dCas9 plasmid was generated by inserting the FKBP12^F36V domain C-terminally to dCas9 into the plasmid pSLQ2818 pPB: CAG-PYL1-KRAB-IRES-Puro-WPRE-SV40PA PGK-ABI-tagBFP-SpdCas9 (Addgene plasmid # 84241; a gift from Stanley Qi), after exchanging the Puromycin resistance with a resistance for Blasticidin.

The hHDAC4 domain used to generate hHDAC4-dCas9 and dCas9-hHDAC4 repression systems was amplified from pEx1-pEF-H2B-mCherry-T2A-rTetR-HDAC4, which was a gift from Michael Elowitz (Addgene plasmid # 78349; RRID:Addgene_78349). Finally, the CasRx (RfxCas13d) coding sequence was amplified from pLentiRNA-CRISPR_007 - TetO-NLS-RfxCas13d-NLS-WPRE-EFS-rtTA3-2A-Blast, which was a gift from Neville Sanjana (Addgene plasmid # 138149; RRID:Addgene_138149). Plasmid maps can be found in the Supplementary Data 4. Plasmids and their associated sequences are deposited at Addgene.

### RNA extraction, reverse transcription, qPCR

To harvest RNA samples for quantitative PCR (qPCR), ~ 2 × 10^6 cells were washed with ice-cold PBS and lysed by directly adding 500 μl of Trizol (Invitrogen). RNA was isolated using the Direct-Zol RNA Miniprep Kit (Zymo Research) following the manufacturer's instructions. 1 μg of RNA were reverse transcribed using Superscript III Reverse Transcriptase (Invitrogen) with random hexamer primers (Thermo Fisher Scientific) and expression levels were quantified in the QuantStudio™ 7 Flex Real-Time PCR machine (Thermo Fisher Scientific) using 2xSybRGreen Master Mix (Applied Biosystems) normalising to Rrm2 and Arpo. All qPCR primers have been validated by PCR and cDNA dilution curve. Primer sequences are listed in Supplementary Data 1 part D.

### Live cell flow cytometry

Cell samples for flow cytometry were harvested by washing with PBS, dissociation with trypsin for 7 min at 37 °C and resuspension in serum-containing medium (DMEM (Sigma), 15% ESC-grade FBS (Gibco), 0.1 mM β-mercaptoethanol). The cell suspension was transferred to a U-bottom 96-well plate, centrifuged for 5 min at 500 × g at 4 °C and resuspended in 70 μl flow cytometry buffer (PBS, 10% ESC-grade FBS (Gibco), 0.5 mM EDTA) on ice. The samples were analysed using BD FACSCelesta Cell Analyzer (Beckton Dickinson, IC-Nr.: 68186, Serial-Nr.: R66034500035) with 2-Blue6-Violet4-561YG laser configuration, equipped with BD High Throughput Sampler (HTS). The HTS option was used for measurement of fluorescence in 96-well plates. tBFP fluorescence was measured using the 450/440 Band Pass filter, tRFP or mCherry using the 586/515 Band Pass filter and EGFP using the 530/30 band pass filter. For the experiments with tRFP and tBFP measurements, violet laser was set to a voltage of 380 and yellow-green laser to 420 V. For the experiments involving mCherry and tBFP measurement, the violet laser was set at 340 and yellow-green at 465 V. For the experiments involving EGFP and tBFP measurements, the violet laser was set at 380 and blue laser at 440 V. When the HTS option was used, the sample flow-rate was set to 1 or 2 μl/s and 70% of the total volume in the well was set at sample volume. At least 10,000 events were recorded. When the cells were analysed using FACS tubes, at least 30,000 events were recorded.

### Flow cytometry data analysis

Data analysis of flow cytometry files was performed using the R programming language and the packages FlowCore, OpenCyto and ggcyto. In order to clearly distinguish events corresponding to single, live cells, two gates are applied sequentially. First, events were gated

based on the Forward and Side Scatter Area (FSC-A and SSC-A) using openCyto:::.boundary() in order to select events corresponding to live cells. Second, to separate single cells (singlets) from doublets, the events showing a linear relationship between Forward Scatter Height and Area (FSC-H and FSC-A) were automatically selected using openCyto:::.singletGate(). Importantly, the same gating coordinates were applied to all files within one experiment and, where possible, across experiments. The experiments involved either tBFP and tRFP as fluorochromes or tBFP and mCherry or tBFP and EGFP. Because the overlap between emission spectra was none or minimal, no compensation was deemed to be necessary.

The fluorescence distributions were plotted by extracting the relevant parameters of the singlet-gated populations using ggplot2 and the ggridges package with the function geom_density_ridges(), which calculates density estimates from the provided data. To quantify protein abundance from flow cytometry data, the Median Fluorescence Intensity (MFI) was calculated for a given sample, followed by correction for the cells' autofluorescence through subtraction of the MFI of one or multiple non-fluorescent control samples of the same cell line. The MFI values and fold changes reported are always after subtraction of background autofluorescence.

### Analysis of degron properties

To analyse the properties associated with the different degrons, each sample was corrected for autofluorescence background by taking the mean of the MFI of non-fluorescent cells treated with the maximal concentration of ligand and mock-treated. We then calculated the tRFP/tBFP ratio and normalised it to the mean of the tRFP/tBFP ratio for the no-degron control cell line, averaged across all replicates treated with maximal ligand concentration and mock-treated samples. This relative tRFP/tBFP ratio was then expressed as a percentage. To quantify degron properties, we then defined the tRFP/tBFP ratio at the stabilised state as A and the ratio under destabilised conditions as B.

The degradation leakiness, which measures background destabilisation of the stabilised state (e.g. destabilisation conferred by the degron tag per se in the absence of degrader), was defined as:

$$100\% - A$$

The degradation efficiency, which measures the reduction of protein abundance relative to the no-degron control, was defined as:

$$100\% - B$$

The dynamic range for each degron-dCas9 construct, which measures the range of protein concentrations that can be achieved by modifying the ligand concentration, is calculated as $B/A$.

### Modelling the dynamics of repression and reversibility

Curve fitting and deterministic modelling were performed using the R programming language and the packages minpack.lm for non-linear least square (NLS) problems computation and deSolve for solving ordinary differential equations.

We described the system with two ordinary differential equations (ODEs), where Eq. (1) describes the expression of the Cas-Repressor R and Eq. (2) the target gene expression T, the production of which is modulated by the Cas-Repressor:

$$dR/dt = \beta_R - \alpha_R \cdot R \tag{1}$$

$$dT/dt = \beta_T(R) - \alpha_T T \tag{2}$$

Where $\alpha$ and $\beta$ denote the degradation and production rates, respectively. $\beta_R$ and $\alpha_T$ are assumed to be constant, while $\alpha_R$ is modulated by

the dTAG-13 concentration and the effective production rate of the target $\beta_T(R)$ is a function of the repressor levels.

**Quantification of repressor half times.** Since $\alpha_R$ is modulated by the dTAG-13 concentration, we estimated the parameter both in the presence ($\alpha_{R+}$) and in the absence ($\alpha_{R-}$) of the ligand from time course data upon dTAG-13 addition and withdrawal, respectively. The background-corrected MFI values were rescaled between 0 and 1. For the dTAG withdrawal time course, where all repressors reached their maximal level in <25 h, the mean MFI of the 25 h–150 h time points was set to 1 and the mean MFI at time 0 h–0. To calculate the time required for the Cas-Repressor to reach half of its maximal concentration, the time course was fit with the analytical solution of Eq. (1) with the initial condition R(0) = 0:

$$R(t)/R_{st} = 1 - e^{-\alpha_R t} \tag{3}$$

Where $R_{st}$ denotes the level of R at steady state (t > 24 h) and $R(t)/R_{st}$ represents the scaled Cas-Repressor expression level. Since the half-time required for the Cas-Repressor to reach the steady state is given by $t_{1/2} = \frac{\ln(2)}{\alpha_{R-}}$, we substituted $\alpha_{R-}$ in Eq. (3) with $\ln(2)/t_{1/2}$ to directly compute the half-time and the associated standard error for each Cas-Repressor. To estimate the repressor degradation rate and associated half time in the presence of dTAG-13, the Cas-Repressor down-regulation time course upon dTAG-13 addition was rescaled such that the mean MFI of the 25–150 h time points, where all repressor constructs had been fully degraded, was set to zero and the mean MFI before the treatment was set to 1. The data was then fit with the analytical solution of Eq. (1) with the initial condition $R_0 = 1$ and a steady state of 0:

$$R(t)/R_0 = e^{-\alpha_{R+} t} \tag{4}$$

Where $\alpha_{R+}$ is the degradation rate of the Cas-Repressor upon dTAG-13 addition and the half time of repressor removal was estimated as described above.

**Parameter estimation of repressor-dependent target repression.** In order to quantify the relationship between Cas-Repressor concentration and target repression $\beta_T = f(R)$, we used the dTAG-13 titration data sets with 4 days of treatment, assuming that the systems had reached a steady state at that time. Background-corrected MFI values of Cas-Repressors (tBFP) were scaled between 0 and 1 (min-max scaling), while the target gene (mCherry) was computed as a fold change relative to levels in the absence of repression (500 nM dTAG-13). Describing the dose-response relationship between the repressor and the target with a Hill function, we then solved Eq. (2) at steady state:

$$\frac{T(R)}{T(R=0)} = \frac{1}{1 + \left(\frac{R}{K}\right)^n} \tag{5}$$

Where $\frac{T(R)}{T(R=0)}$ describes the scaled target levels, K denotes the levels of R that leads to 50% target repression and n represents the Hill coefficient. K and n were estimated for each repressor construct.

**Estimation of the target degradation rate.** To estimate the degradation rate of the target gene $\alpha_T$, we used the mCherry time course upon ligand addition in the CasRx cell line. Since the repressor was fully degraded in <1 h and target repression can be assumed to be immediate in this system, which relies on mRNA degradation, the dynamics of reporter upregulation can be assumed to be uniquely determined by the mCherry degradation rate. After normalising the background-subtracted MFI of the target gene to the mean MFI at time point 150 h, we estimated the mCherry degradation rate by fitting the

analytical solution of Eq. (2) in the absence of repression to the data:

$$T(t)/T_{st} = 1 + (T_0 - 1)e^{-\alpha_T t} \tag{6}$$

Where $T_{st}$ is the final target level (at 150 h) and $T_0$ the initial target level. The mCherry degradation rate $\alpha_T$, was estimated to be $0.036\,h^{-1}$.

**Quantification of repression delay.** To understand whether target repression and derepression is immediate or follows changes in repressor levels with a delay, we used the ODE model to simulate target expression with the estimated parameters, either assuming an immediate ($\Delta t = 0$) or a delayed ($\Delta t > 0$) effect of the repressor on the target. Different values were tested for $\Delta t$, ranging from 0 to 25 h and numerical solutions were compared to the experimentally measured mCherry levels through computing the Mean Absolute Error (MAE). The $\Delta t$ that minimised the MAE is reported.

**Statistical analysis of distribution modality**
To assess the modality of ESRRB-mCherry distributions and STAG2-EGFP distributions at different dTAG-13 concentrations (Fig. 3 and Fig. 5), the method proposed by Hall and York[67], implemented in the R package multimode, was used (function modetest(method = "HY", submethod = 1) with default parameters). The method computes the critical (i.e. smallest) bandwidth required to obtain a kernel density estimation of the experimental data with a single mode. Larger critical bandwidths correspond to less unimodal distributions. The experimental distributions were log10-transformed prior to the test and the three replicates for each dTAG-13 concentration merged. The possible locations of the modes were bound between the 5th and 95th percentile of each merged distribution, to avoid detection of spurious modes along the tails.

**Modelling of NANOG and OCT4 dose-response curves**
To quantify how NANOG or OCT4 dose relates to variation in gene expression of specific genes, we fitted a Hill function to the dose-response curves. To this end, target gene expression was first normalised to the not-repressed state (500 nM dTAG-13). Then we used the minpack.lm package in R, to fit the dose-response curve with a 4 parameter Hill-type function using a non-linear least square approach using the formula:

$$Target\,FC = T_{min} + \frac{T_{max} - T_{min}}{1 + \left(\frac{[X]1/2}{X}\right)^n}$$

The equation describes the fold change variation of the target ($Target\,FC$) as a function of Nanog or Oct4 fold change ($X$). $T_{max}$ is the target fold change at 500 nM dTAG-13 and is set equal to 1, while $T_{min}$ is the target fold change when Nanog or Oct4 is 0 and is estimated by fitting the data with the NLS approach. Through this, we can extract $[N]_{1/2}$ or $[O]_{1/2}$, which corresponds to the fold change of Nanog or Oct4, respectively, that leads to half of the maximal target variation, and the Hill coefficient $n$, which indicates the degree of non-linearity. The same equation was also used to estimate how NANOG dose relates to the differentiation status: the percentage of undifferentiated colonies from the alkaline phosphatase assay was normalised to the percentage in the 500 nM dTAG-13 condition and for NANOG we calculated the fold change of the background-subtracted MFI measured by flow-cytometry prior to seeding the cells for the clonogenic assay.

Lastly, for genes repressed by NANOG or OCT4, the equation is written as:

$$Target\,FC = T_{max} - \frac{T_{max} - T_{min}}{1 + \left(\frac{[X]1/2}{X}\right)^n}$$

with $T_{min}$ set equal to 1.

## Statistics and reproducibility
No statistical method was used to predetermine the sample size. No data were excluded from the analysis. The experiments were not randomised. The investigators were not blinded to allocation during experiments and outcome assessment.

## Reporting summary
Further information on research design is available in the Nature Portfolio Reporting Summary linked to this article.

## Data availability
All raw flow cytometry data generated in this study are available on request due to the large file size, requests should be made to the corresponding author and will be answered within 2 weeks. All processed data presented in this study are provided as a Source Data file. Plasmids are available through Addgene. A minimal dataset has been deposited at Github [https://github.com/EddaSchulz/CasTuner/], with https://doi.org/10.5281/zenodo.7928748.

## Code availability
Original code to estimate repression and derepression dynamics has been deposited at https://github.com/EddaSchulz/CasTuner/, with https://doi.org/10.5281/zenodo.7928748.

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

## Acknowledgements

We thank Oriana Genolet for generating the 1.8XX Esrrb-mCherry and Nanog-mCherry cell lines. We also thank Liat Ravid Lustig for scientific advice and Francesca Rossi for scientific discussion. We thank Tugce Aktas and Sarah Kinkley for sharing reagents. We thank Michael Böttcher and Ghanem El Kassem for advice on CasRx usage. We thank Jan-Michael Peters (IMP Vienna) for sharing the STAG2-EGFP HeLa cell line. We thank Till Schwämmle for cloning the Oct4 CRISPRi guides. We thank the Max Planck Institute for Molecular Genetics FACS facility. This work was supported by the Max-Planck Research Group Leader program, E:bio Module III—Xnet grant (BMBF 031L0072), Human Frontiers Science Program (CDA-00064/2018) and ERC Starting Grant CisTune (948771) to E.G.S. G.N. was supported by the European Union's Horizon 2020 Research and Innovation Program (Marie Skłodowska-Curie ITN PEP-NET).

## Author contributions

G.N. and E.G.S. conceived the project and designed the experiments. R.A.F.G. designed and cloned most of the degron-dCas9-tRFP-P2A-tBFP plasmids. G.N. designed and cloned all other plasmids, performed all the experiments and analysed the data. G.N. and E.G.S. wrote the paper with inputs from R.A.F.G.

## Funding

## Competing interests

The authors declare no competing interests.
