## [Peer Review File · Nature Communications]

Reviewers' Comments:

Reviewer #1:

Remarks to the Author:

Tuning the expression level of endogenous genes is potentially powerful for dissecting biological networks within cells. Noviello et al. developed a new method called CasTuner that allows downregulating the expression level of endogenous genes using a degron-fused repressor or RNA-targeting Cas (CasRx). Initially, the authors explored choosing the best ligand-dependent degron system for controlling dCas9 (Fig. 1). By looking at the leakiness, degradation efficiency and dynamic range, they found that the dTAG system worked the best among the tested degrons. Subsequently, they fused dTAG and Cas9 to KRAB, hHDAC4, or CasRx to control the endogenous *Esrrb1* gene (Fig. 2). In mESC expressing *Esrrb1* targeting gRNAs, *ESRRB1*/mCherry was induced or suppressed with or without 500 nM dTAG-13, respectively. The cells expressing dTAG-fused hHDAC4-dCas9 and CasRx showed intermediate *ESRRB1* levels when treated with differential dTAG-13 concentrations (Fig. 3). On the other hand, the KRAB-dCas9 expressing cells showed a bimodal change. They looked at the repression and induction kinetics of *ESRRB1* tuned by these systems (Fig. 4) and found that KRAB-dCas9 showed a delayed response. Finally, the authors controlled the *Nanog* gene using hHDAC4-Cas9 (Fig. 5). They found that the downstream genes responded with a differential *NANOG* level, showing that each downstream gene has a differential *NANOG* threshold.

This is a comprehensive study done with a neat experimental setup, and the data quality presented in this MS is high. I agree with the authors' conclusion that CasTuner is a new tool to perturb endogenous gene expression in an inducible, tunable and reversible manner. However, considering this is a methodology paper, I have some concerns regarding the generality and safety of the system.

One of my major concerns is the generality of CasTuner. Because the authors used only mESC, I am curious to know if CasTuner can be used in human cell lines.

CasRx (also known as Cas13d) is reported to degrade non-target RNAs when activated by guide-RNA (PMID: 29551272). Multiple recent papers showed that this collateral activity caused cytotoxicity and lethality in mice (PMID: 31637166, 35244715; BioRxiv: 10.1101/2021.12.20.473384, 10.1101/2021.11.30.470032, 10.1101/2022.01.17.476700), pointing out the danger of using CasRx. To overcome this problem, recent papers reported a Cas13d variant lacking collateral activity and a new Cas without this activity (PMID: 34489594, 35953673). I wonder if the authors observed cytotoxicity or changes in transcriptome when CasRx was expressed with non-targeting control (NTC) gRNAs. If so, I highly suggest using a safer Cas13 variant than the original CasRx.

How completely can the authors suppress the expression of *ESRRB1* using the hHDAC4-dCas9 and CasRx systems? Fig. 2g shows that its expression was almost 100% induced when the cells were treated with 500 nM dTAG-13. However, the authors do not show how tightly the *ESRRB1* expression is suppressed in the absence of dTAG-13. This is because the authors did not take a baseline control using mESC without mCherry and do not set the 0% level. I also noted that there was considerable cell-to-cell variability of the mCherry expression levels when treated with 0 nM dTAG-13 (Fig. 3C and Fig. S3C). This point should be stated and discussed in the MS.

Reviewer #2:

Remarks to the Author:

The manuscript by Gemma Noviello et al. provides and characterizes a toolkit for finetuning CRISPR/dCas9 based repression. Degrons that are fused to dCas9 act as conditional destabilizing domains that can be fine-tuned by ligand concentration. After initially testing six degrons, they focus on the FKBP12[F36V] degron domain which demonstrated a wide dynamic range of repression (tunability). Tunable CRISPR based downregulation is then characterized for two degron-Cas-repression systems (CasTuner): 1) HDAC4-dCas9 acts by deacetylating target DNA thereby repressing transcription, while 2) the RNA-targeting CasRx acts on the post transcriptional

level. Using the endogenously tagged *Esrrb* gene, repression and derepression dynamics were examined and highlight differences between the individual repression systems, contrasting tunable repressors with the widely used CRISPR/dCas9-KRAB system. CasTuner was then used to study the responsiveness of Nanog target genes to varying Nanog levels. This example nicely demonstrates the use of CasTuner to study dose-dependent cellular responses.

Tunability is an important aspect of engineering transcriptional regulation and hence would be of broad interest.

Minor comments:

Was there an unbiased statistical method used to determine unimodal and bimodal distribution? If not, an unbiased method should be applied.

Section 'Speed and reversibility of repression in degron-Cas-repressor systems' should at least mention once that dynamics were evaluated at the target locus was *ESRRB*-mCherry and what cell type.

P13, last paragraph: hHDAC4 should be hHDAC4-dCas9

Could you please be more specific about which repression was immediate (p13, last paragraph). For example, I assume it was the target gene: "Once the repressor was upregulated, repression was immediate for dCas9, hHDAC4-dCas9 and CasRx...".

Reviewer #3:

Remarks to the Author:

The study by Noviello et al. describes the development of the so-called CasTuner technique, based on a degron and CRISPR/Cas-based toolkit, for analog tuning of endogenous gene expression. The authors started by testing six different degron domains (AID, mAID, SMASh, FKBP12-F36V, ecDHFR, and ER50) and narrowed down to the AID and FKBP12-F36V degron domains as being the most potent in controlling the abundance of dCas9 with a wide dynamic range, high degradation efficiency, and intermediate degradation leakiness. The authors then focus on the FKBP12-F36V system, which has a slightly higher folder change and a lower degradation leakiness than the AID system, to demonstrate its utility in perturbing gene expression in an inducible, tunable, and reversible manner and studying dose-responsive processes within the physiological context, using pluripotency-associated genes *Esrrb* and *Nanog* as examples.

The manuscript is in general clearly written, and the technique could potentially be helpful to fine-tune gene expression with an easy-to-implement tool. However, I have some concerns regarding the validity of the CasTuner system presented by the authors.

1. The self-renewal and differentiation status of mouse ESCs is directly responsive to the dose-dependent expression of *Nanog* (Chambers et al., 2007). Although the authors presented the dose-response curves of *NANOG* measured using CasTuner, it is unclear whether that is relevant to the differentiation status of the cells being tested. Therefore, the functional readout of these cells (undifferentiated, partially differentiated, and fully differentiated) should be presented side by side with the *NANOG* doses to support the validity/utility of this CasTuner system.
2. *Esrrb* is the primary direct target gene of *NANOG* in mESCs (Festuccia et al., 2012). In their test of CasTuner to quantify the dose-response curves between *NANOG* and its target genes, *Esrrb* is missing and should be included.
3. Hitoshi Niwa has utilized a genetic approach that demonstrated quantitative expression of Oct-3/4 in defining differentiation, dedifferentiation, or self-renewal of ES cells (Niwa et al., 2000). Specifically, they showed that to maintain the undifferentiated stem cell phenotype, Oct-3/4 expression must remain within plus or minus 50% of normal diploid expression. Therefore, I think Oct-3/4 could be the best candidate to show the utility of this new CasTuner system with a direct comparison with the well-established dose-dependent effect of Oct-3/4 in the ESC control.

References:

Chambers, I., Silva, J., Colby, D., Nichols, J., Nijmeijer, B., Robertson, M., Vrana, J., Jones, K., Grotewold, L., and Smith, A. (2007). Nanog safeguards pluripotency and mediates germline development. *Nature* 450, 1230-1234. 10.1038/nature06403.

Festuccia, N., Osorno, R., Halbritter, F., Karwacki-Neisius, V., Navarro, P., Colby, D., Wong, F., Yates, A., Tomlinson, S.R., and Chambers, I. (2012). Esrrb is a direct Nanog target gene that can substitute for Nanog function in pluripotent cells. *Cell Stem Cell* 11, 477-490. 10.1016/j.stem.2012.08.002.

Niwa, H., Miyazaki, J., and Smith, A.G. (2000). Quantitative expression of Oct-3/4 defines differentiation, dedifferentiation or self-renewal of ES cells. *Nat Genet* 24, 372-376.

ANSWERS TO THE REVIEWERS COMMENTS (Noviello et al.)

We would like to thank all the reviewers for taking the time to read our manuscript. We honestly appreciated the comments, which highlighted the thorough attention to the content, and we tried our best to fulfil them. We have added several new data sets and analyses, including a test of CasTuner in HeLa cells, a phenotypic readout upon Nanog titration, a dose-response analysis for Oct4 and a statistical analysis of bimodality.

Reviewer #1:

Tuning the expression level of endogenous genes is potentially powerful for dissecting biological networks within cells. Noviello et al. developed a new method called CasTuner that allows downregulating the expression level of endogenous genes using a degron-fused repressor or RNA-targeting Cas (CasRx). Initially, the authors explored choosing the best ligand-dependent degron system for controlling dCas9 (Fig. 1). By looking at the leakiness, degradation efficiency and dynamic range, they found that the dTAG system worked the best among the tested degrons. Subsequently, they fused dTAG and Cas9 to KRAB, hHDAC4, or CasRx to control the endogenous *Esrrb1* gene (Fig. 2). In mESC expressing *Esrrb1* targeting gRNAs, *ESRRB1/mCherry* was induced or suppressed with or without 500 nM dTAG-13, respectively. The cells expressing dTAG-fused hHDAC4-dCas9 and CasRx showed intermediate *ESRRB1* levels when treated with differential dTAG-13 concentrations (Fig. 3). On the other hand, the KRAB-dCas9 expressing cells showed a bimodal change. They looked at the repression and induction kinetics of *ESRRB1* tuned by these systems (Fig. 4) and found that KRAB-dCas9 showed a delayed response. Finally, the authors controlled the *Nanog* gene using hHDAC4-Cas9 (Fig. 5). They found that the downstream genes responded with a differential *NANOG* level, showing that each downstream gene has a differential *NANOG* threshold.

This is a comprehensive study done with a neat experimental setup, and the data quality presented in this MS is high. I agree with the authors' conclusion that CasTuner is a new tool to perturb endogenous gene expression in an inducible, tunable and reversible manner. However, considering this is a methodology paper, I have some concerns regarding the generality and safety of the system.

1. One of my major concerns is the generality of CasTuner. Because the authors used only mESC, I am curious to know if CasTuner can be used in human cell lines.

We want to thank the reviewer for prompting us to test the CasTuner system in a different cellular context. In the revised version of the manuscript we demonstrate CasTuner's applicability in human HeLa cells, taking advantage of a previously generated cell line where the endogenous *STAG2* gene is homozygously tagged with EGFP. The new data proved in Fig. 5 and Supplementary Fig. 5 show that the two CasTuner systems allow tunable repression also in this cellular context, while KRAB-mediated repression is again digital.

2. CasRx (also known as Cas13d) is reported to degrade non-target RNAs when activated by guide-RNA (PMID: 29551272). Multiple recent papers showed that this collateral activity caused cytotoxicity and lethality in mice (PMID: 31637166, 35244715; BioRxiv: 10.1101/2021.12.20.473384, 10.1101/2021.11.30.470032, 10.1101/2022.01.17.476700), pointing out the danger of using CasRx. To overcome this problem, recent papers reported a Cas13d variant lacking collateral activity and a new Cas without this activity (PMID: 34489594, 35953673). I wonder if the authors observed cytotoxicity or changes in transcriptome when CasRx was expressed with non-targeting control (NTC) gRNAs. If so, I highly suggest using a safer Cas13 variant than the original CasRx.

As pointed out by the reviewer, recent data indeed suggests that CasRx might exhibit unwanted side effects. When analysing *Esrrb* expression, which is also a marker for the pluripotent state, we do not detect any obvious effects in CasRx-expressing cells with NTC gRNAs (Supplementary Fig. 3a-b). We have not assessed potential collateral effects in a transcriptome-wide manner, as it would go beyond the scope of our work, which mainly focuses on the tunability of the system through a degron domain. However, we now discuss the reported collateral effects and the possibility to substitute the CasRx domain used in our toolkit with one of the recently described mutagenised variants with less or no collateral activity (p. 24).

3. How completely can the authors suppress the expression of *ESRRB1* using the hHDAC4-dCas9 and CasRx systems? Fig. 2g shows that its expression was almost 100% induced when the cells were treated with 500 nM dTAG-13. However, the authors do not show how tightly the *ESRRB1* expression is suppressed in the absence of dTAG-13. This is because the authors did not take a baseline control using mESC without mCherry and do not set the 0% level.

We apologise that the quantification of the flow cytometry data was not described with sufficient clarity in the original version of the manuscript. For all comparisons we first subtract the background fluorescence of a non-fluorescent control cell line. The quantification in Fig. 2f thus shows that the repression is 80%-90%, since the MFI ratio between -dTAG (repressed) and +dTAG (not repressed) lies around 0.1-0.2. We have modified the axis labels and figure legends to make this clear and we have added the non-fluorescent control to the density plots in Fig. 2c+e.

4. I also noted that there was considerable cell-to-cell variability of the mCherry expression levels when treated with 0 nM dTAG-13 (Fig. 3C and Fig. S3C). This point should be stated and discussed in the MS.

This point is discussed now on p.12 of the revised version of the manuscript.

Reviewer #2:

The manuscript by Gemma Noviello et al. provides and characterizes a toolkit for finetuning CRISPR/dCas9 based repression. Degrons that are fused to dCas9 act as conditional destabilizing domains that can be fine-tuned by ligand concentration. After initially testing six degrons, they focus on the FKBP12[F36V] degnon domain which demonstrated a wide dynamic range of repression (tunability). Tunable CRISPR based downregulation is then characterized for two degnon-Cas-repression systems (CasTuner): 1) HDAC4-dCas9 acts by deacetylating target DNA thereby repressing transcription, while 2) the RNA-targeting CasRx acts on the post transcriptional level. Using the endogenously tagged Esrrb gene, repression and derepression dynamics were examined and highlight differences between the individual repression systems, contrasting tunable repressors with the widely used CRISPR/dCas9-KRAB system. CasTuner was then used to study the responsiveness of Nanog target genes to varying Nanog levels. This example nicely demonstrates the use of CasTuner to study dose-dependent cellular responses.

Tunability is an important aspect of engineering transcriptional regulation and hence would be of broad interest.

Minor comments:

1. Was there an unbiased statistical method used to determine unimodal and bimodal distribution? If not, an unbiased method should be applied.

For a more unbiased comparison of the distributions observed upon targeting with the different Cas-repressor systems, we have performed a statistical analysis according to the method proposed by Hall and York. It computes a statistic called “critical bandwidth”, which corresponds to the kernel density of the distribution required to render a distribution unimodal. The associated test has the null-hypothesis of a distribution being unimodal. We applied this test to the Esrrb-mCherry data in mESCs (Supplementary Fig. 3f) and to the newly generated data for STAG2-GFP in HeLa cells (Supplementary Fig. 5c). In both cases we observed an increased critical bandwidth and $p < 0.01$ only for the KRAB systems, supporting our conclusion that those systems exhibit pronounced bimodality.

2. Section ‘Speed and reversibility of repression in degnon-Cas-repressor systems’ should at least mention once that dynamics were evaluated at the target locus was ESRRB-mCherry and what cell type.

We modified the text as suggested (p. 13).

3. P13, last paragraph: hHDAC4 should be hHDAC4-dCas9

This has been corrected.

4. Could you please be more specific about which repression was immediate (p13, last paragraph). For example, I assume it was the target gene: “Once the repressor was upregulated, repression was immediate for dCas9, hHDAC4-dCas9 and CasRx...”.

We have restructured this sentence to improve clarity (p. 14, top).

Reviewer #3:

The study by Noviello et al. describes the development of the so-called CasTuner technique, based on a degron and CRISPR/Cas-based toolkit, for analog tuning of endogenous gene expression. The authors started by testing six different degron domains (AID, mAID, SMASh, FKBP12-F36V, ecDHFR, and ER50) and narrowed down to the AID and FKBP12-F36V degron domains as being the most potent in controlling the abundance of dCas9 with a wide dynamic range, high degradation efficiency, and intermediate degradation leakiness. The authors then focus on the FKBP12-F36V system, which has a slightly higher fold change and a lower degradation leakiness than the AID system, to demonstrate its utility in perturbing gene expression in an inducible, tunable, and reversible manner and studying dose-responsive processes within the physiological context, using pluripotency-associated genes *Esrrb* and *Nanog* as examples.

The manuscript is in general clearly written, and the technique could potentially be helpful to fine-tune gene expression with an easy-to-implement tool. However, I have some concerns regarding the validity of the CasTuner system presented by the authors.

1. The self-renewal and differentiation status of mouse ESCs is directly responsive to the dose-dependent expression of *Nanog* (Chambers et al., 2007). Although the authors presented the dose-response curves of *NANOG* measured using CasTuner, it is unclear whether that is relevant to the differentiation status of the cells being tested. Therefore, the functional readout of these cells (undifferentiated, partially differentiated, and fully differentiated) should be presented side by side with the *NANOG* doses to support the validity/utility of this CasTuner system.

We want to thank the reviewer for this valuable suggestion to include a phenotypic readout in the *Nanog* titration experiment. We have now performed a colony growth assay with alkaline phosphatase staining upon *Nanog* titration. The new data are reported in Fig. 6e-g and discussed in the text at p. 19.

2. *Esrrb* is the primary direct target gene of *NANOG* in mESCs (Festuccia et al., 2012). In their test of CasTuner to quantify the dose-response curves between *NANOG* and its target genes, *Esrrb* is missing and should be included.

Esrrb has now been included in the analysis (Fig. 6d).

3. Hitoshi Niwa has utilized a genetic approach that demonstrated quantitative expression of Oct-3/4 in defining differentiation, dedifferentiation, or self-renewal of ES cells (Niwa et al., 2000). Specifically, they showed that to maintain the undifferentiated stem cell phenotype, Oct-3/4 expression must remain within plus or minus 50% of normal diploid expression. Therefore, I think Oct-3/4 could be the best candidate to show the utility of this new CasTuner system with a direct comparison with the well-established dose-dependent effect of Oct-3/4 in the ESC control.

We were really happy to follow this suggestion, since Oct4 is indeed a striking example of a transcription factor with well characterised and interesting dose-dependent effects. We have titrated Oct4 in our 1.8XX Nanog-mCherry mESC line and quantified a panel of marker genes of the lineages in the early embryo. This new data is reported in Fig. 7 and in Supplementary Fig. 7 and discussed on p. 20. As reported by Niwa et al., at a low Oct4 dose we observe upregulation of trophectodermal genes, when Oct4 drops below 30-50% of wildtype levels. Interestingly, we observed an increase in Nanog expression at intermediate Oct4 levels (~50%), associated with a more naïve-like morphology, which resembles the phenotype reported previously for Oct4^{+/-} mESCs (Karwacki-Neisius et al., 2013, Radziskeuskaya et al, 2013).

Reviewers' Comments:

Reviewer #1:

Remarks to the Author:

My concerns have been addressed. I believe that the manuscript is now ready to be accepted.

A typo

"histone 3 lysine 9 trimethylation" should be " histone H3 lysine 9 trimethylation" (page 7).

Reviewer #2:

Remarks to the Author:

The authors have addressed all of my concerns.

Reviewer #3:

Remarks to the Author:

The authors have done a good job addressing my initial concerns and those raised by other reviewers. The inclusion of Nanog tuning effects on clonogenicity phenotypic readout, the Oct4 CasTuner, and the application of the CasTuner system to human cells makes this manuscript more attractive with greater impact. I recommend it for publication in a timely manner.

Point-by-point response

Reviewer #1

My concerns have been addressed. I believe that the manuscript is now ready to be accepted.

A typo

"histone 3 lysine 9 trimethylation" should be " histone H3 lysine 9 trimethylation" (page 7).

The typo has been corrected.

Reviewer #2 (Remarks to the Author):

The authors have addressed all of my concerns.

Reviewer #3 (Remarks to the Author):

The authors have done a good job addressing my initial concerns and those raised by other reviewers. The inclusion of Nanog tuning effects on clonogenicity phenotypic readout, the Oct4 CasTuner, and the application of the CasTuner system to human cells makes this manuscript more attractive with greater impact. I recommend it for publication in a timely manner.